# Faster Wasserstein Distance Estimation with the Sinkhorn Divergence

**Lénaïc Chizat**[1*],    **Pierre Roussillon**[2],    **Flavien Léger**[2],
**François-Xavier Vialard**[3],    **Gabriel Peyré**[2]

1: Laboratoire de Mathématiques d'Orsay, CNRS, Université Paris-Saclay, Orsay, France
2: ENS, PSL University, Paris, France
3: Univ. Gustave Eiffel, CNRS, ESIEE Paris, Marne-la-Vallée, France

## Abstract

The squared Wasserstein distance is a natural quantity to compare probability distributions in a non-parametric setting. This quantity is usually estimated with the plug-in estimator, defined via a discrete optimal transport problem which can be solved to $\varepsilon$-accuracy by adding an entropic regularization of order $\varepsilon$ and using for instance Sinkhorn's algorithm. In this work, we propose instead to estimate it with the Sinkhorn divergence, which is also built on entropic regularization but includes debiasing terms. We show that, for smooth densities, this estimator has a comparable sample complexity but allows higher regularization levels, of order $\varepsilon^{1/2}$, which leads to improved computational complexity bounds and a strong speedup in practice. Our theoretical analysis covers the case of both randomly sampled densities and deterministic discretizations on uniform grids. We also propose and analyze an estimator based on Richardson extrapolation of the Sinkhorn divergence which enjoys improved statistical and computational efficiency guarantees, under a condition on the regularity of the approximation error, which is in particular satisfied for Gaussian densities. We finally demonstrate the efficiency of the proposed estimators with numerical experiments.

## 1 Introduction

Certain tasks in machine learning (implicit generative modeling [41], two-sample testing [50], structured prediction [25]) and imaging sciences (shape matching [31], computer graphics [9]) require to quantify how much two probability densities $\mu, \nu \in \mathcal{P}(\mathbb{R}^d)$ differ. The squared Wasserstein distance $W_2^2(\mu, \nu)$ (defined below) is often well suited for this purpose because of its appealing geometrical properties [57, 52, 46] but it also raises important statistical and computational challenges. Indeed, in many practical settings, $\mu$ and $\nu$ are only accessed via empirical or discretized measures $\hat{\mu}_n, \hat{\nu}_n$ composed of $n$ atoms. A standard workaround is to use the *plug-in estimator* $W_2^2(\hat{\mu}_n, \hat{\nu}_n)$, but although it is efficient when $\mu$ and $\nu$ are discrete [55, 56], this estimator suffers from the curse of dimensionality when $\mu$ and $\nu$ have densities [59, Cor. 2], with an estimation error that scales as $n^{-2/d}$ as we show in Section 3. Moreover, solving the discrete optimal transport problem is computationally demanding when $n$ is large, with a time complexity bound scaling as $n^2 \log(n)/\varepsilon^2$ to reach $\varepsilon$-accuracy with Sinkhorn's algorithm [20, 2]. These drawbacks give a strong motivation to define and study alternative estimators for $W_2^2(\mu, \nu)$ when $\mu$ and $\nu$ admit smooth densities.

**Entropic regularization of optimal transport.** In this paper, we consider instead estimators based on the idea of entropic regularization of optimal transport [61, 21, 36, 16]. When $\mu$ and $\nu$ have finite

---

[*]Corresponding author: `lenaic.chizat@universite-paris-saclay.fr`

second moments, the entropy regularized optimal transport cost is defined as

$$T_\lambda(\mu, \nu) \stackrel{\text{def.}}{=} \min_{\gamma \in \Pi(\mu,\nu)} \int_{(\mathbb{R}^d)^2} \|y - x\|_2^2 \, \mathrm{d}\gamma(x,y) + 2\lambda H(\gamma, \mu \otimes \nu) \tag{1}$$

where $\Pi(\mu, \nu)$ is the set of transport plans between $\mu$ and $\nu$, $\lambda \geq 0$ is the regularization parameter, and $H(\gamma, \mu \otimes \nu)$ is the entropy of $\gamma$ with respect to the product measure $\mu \otimes \nu$ (see details in the Notations paragraph). The squared Wasserstein distance is defined as $W_2^2(\mu, \nu) \stackrel{\text{def.}}{=} T_0(\mu, \nu)$. Entropic regularization has been popularized as a method to compute $W_2^2(\hat{\mu}_n, \hat{\nu}_n)$ efficiently or simply as a different notion of discrepancy between measures. In contrast, we use it as a tool to directly estimate $W_2^2(\mu, \nu)$. For this purpose, the choice $T_\lambda(\hat{\mu}_n, \hat{\nu}_n)$ is not ideal because its large bias requires to set $\lambda$ to a small value, leading to computational difficulties.

**The proposed estimators.** The first estimator that we consider is $\hat{S}_{\lambda,n} = S_\lambda(\hat{\mu}_n, \hat{\nu}_n)$ where $S_\lambda$ is the *Sinkhorn divergence* [50] defined as

$$S_\lambda(\mu, \nu) \stackrel{\text{def.}}{=} T_\lambda(\mu, \nu) - \frac{1}{2}\big(T_\lambda(\mu, \mu) + T_\lambda(\nu, \nu)\big). \tag{2}$$

In previous work [23], the debiasing terms have been theoretically justified as a mean to have $S_\lambda(\mu, \nu) \geq 0$ with equality when $\mu = \nu$, a property not satisfied by $T_\lambda$. In the present work, we show that they in fact allow, under regularity assumptions, to approximate $W_2^2(\mu, \nu)$ with an error of order $\lambda^2$, instead of $\lambda \log(1/\lambda)$ for the uncorrected quantity $T_\lambda$. We also consider the estimator $\hat{R}_{\lambda,n} = R_\lambda(\hat{\mu}_n, \hat{\nu}_n)$ where $R_\lambda$ is built from $S_\lambda$ via Richardson extrapolation as

$$R_\lambda(\mu, \nu) \stackrel{\text{def.}}{=} 2S_\lambda(\mu, \nu) - S_{\sqrt{2}\lambda}(\mu, \nu). \tag{3}$$

This estimator has a smaller approximation error in $o(\lambda^2)$ and potentially in $O(\lambda^4)$ under restrictive regularity assumptions.

**Contributions.** We make the following contributions:

- In Section 2, we exploit the dynamical formulation of (1) to show that $|S_\lambda(\mu, \nu) - W_2^2(\mu, \nu)| \leq \lambda^2 I$ where $I$ depends on the Fisher information of $\mu$, of $\nu$ and of the $W_2$-geodesic connecting them. We also give a second-order expansion of this approximation error and detail several situations where $I$ admits a priori bounds.

- In Section 3.1, we prove a sample complexity bound for the plug-in estimator $W_2^2(\hat{\mu}_n, \hat{\nu}_n)$ of order $n^{-2/d}$ which has a tight exponent in contrast to the previously known rate $n^{-1/d}$. This is the baseline rate against which we compare the performance of $\hat{S}_{\lambda,n}$ and of $\hat{R}_{\lambda,n}$.

- In Section 3.2, we study the performance of the Sinkhorn divergence estimator $\hat{S}_{\lambda,n}$ given independent samples. We show that when $\lambda$ is properly chosen, it enjoys comparable sample complexity bounds and improved computational guarantees in a certain sense. We also study the performance when the marginals are discretized on a uniform grid in Section 3.3.

- In Section 4, we study estimators based on Richardson extrapolation such as $\hat{R}_{\lambda,n}$. Under an abstract and stronger regularity assumption, this estimator enjoys better computational and sample complexity bounds than the plug-in estimator. We discuss this assumption and show that it is satisfied for Gaussian densities.

- In Section 5, we perform numerical experiments that confirm the benefits of the proposed estimators and suggest that our theoretical results could be extended in several ways.

**Previous Works.** Without additional assumptions, no estimator achieves better statistical rates than the plug-in estimator [44, Thm. 3]. Recent breakthroughs in statistical optimal transport [60, 33] have shown that other estimators can exploit smoothness assumptions to attain faster and nearly minimax estimation rates for $W_2$ or the dual potentials, but they are *a priori* not computationally efficient. In contrast, our goal in this paper is to improve the computational efficiency of estimating $W_2^2(\mu, \nu)$ and we are not aiming at statistical optimality.

The idea of entropic regularization has a long history in computational optimal transport. It has been shown in [2, 20] that solving $T_\lambda(\hat{\mu}_n, \hat{\nu}_n)$ to $\varepsilon$-accuracy requires $O(n^2/(\lambda\varepsilon))$ arithmetic operations

using Sinkhorn's algorithm if the domain is bounded (see Appendix B). We use this bound in our discussions on computational complexity because it cleanly quantifies how harder the problem becomes as $\lambda$ becomes smaller and also because Sinkhorn's algorithm is simple to implement and widely used in practice. Choosing $\lambda \asymp \varepsilon / \log(n)$ allows in turn to estimate $W_2^2(\hat{\mu}_n, \hat{\nu}_n)$ to $\varepsilon$-accuracy in $O(n^2 \log(n)/\varepsilon^2)$ operations [20]. There are however various algorithms with better guarantees both for the regularized [20, 1, 13] and the unregularized problem [37, 47, 7]. In our numerical experiments, we use Sinkhorn's iterations combined with Anderson's acceleration [3, 53], which in practice strongly speeds up convergence.

In front of the difficulty to estimate $W_2^2(\mu, \nu)$, researchers have also turned their attention to similar but more tractable discrepancy measures such as the sliced Wasserstein distance [49] or the Sinkhorn divergence [50], which can be both estimated at the parametric rate [26, 40, 39, 42]. However, there is "no free lunch" and unconditional statistical efficiency comes at the price of lack of adaptivity and discriminative power. In particular, it is known that when $\lambda \to \infty$, $S_\lambda(\mu, \nu)$ converges to the squared distance between the expectations of $\mu$ and $\nu$, which is a degenerate form of Kernel Mean Discrepancy [27, 23]. This shows that the discriminative power of $S_\lambda$ decreases as $\lambda$ increases, but this phenomenon is not yet well understood nor quantified. From a theoretical viewpoint, we thus believe that seeing $S_\lambda$ as an estimator for $W_2^2$ allows to clarify the trade-offs at play in the choice of $\lambda$ between the statistical, approximation and computational errors.

**Notations.** For two probability measures $\mu, \nu \in \mathcal{P}(\mathbb{R}^d)$, we denote by $\Pi(\mu, \nu)$ the set of transport plans between $\mu$ and $\nu$, which is the set of measures $\gamma \in \mathcal{P}(\mathbb{R}^d \times \mathbb{R}^d)$ with marginal $\mu$ (resp. $\nu$) on the first (resp. second) factor of $\mathbb{R}^d \times \mathbb{R}^d$. The quantity $H(\mu, \nu)$ is the entropy of $\mu$ relative to $\nu$, defined as $H(\mu, \nu) \overset{\text{def.}}{=} \int \log(d\mu/d\nu)d\mu$ when $\mu$ is absolutely continuous with respect to $\nu$, and $+\infty$ otherwise. When $\mu$ has a density with respect to the Lebesgue measure, written $\mu(x)$, we define $H(\mu) \overset{\text{def.}}{=} \int \log(\mu(x))\mu(x)dx$ its entropy relative to the Lebesgue measure. Finally, $\mu \otimes \nu \in \mathcal{P}(\mathbb{R}^d \times \mathbb{R}^d)$ is the product measure characterized by $(\mu \otimes \nu)(A \times B) = \mu(A)\nu(B)$ for any pair of Borel sets $A, B \subset \mathbb{R}^d$.

## 2 Refined approximation bound for the Sinkhorn divergence

In this section, we study the approximation error of $S_\lambda$. To this goal, we leverage the dynamical formulation of entropic optimal transport [12, 28, 30, 14] which states that, for $\mu, \nu \in \mathcal{P}(\mathbb{R}^d)$ absolutely continuous probability measures with compact support,

$$T_\lambda(\mu, \nu) + d\lambda \log(2\pi\lambda) + \lambda(H(\mu) + H(\nu)) =$$
$$\inf_{\rho, v} \int_0^1 \int_{\mathbb{R}^d} \left( \|v(t, x)\|_2^2 + \frac{\lambda^2}{4} \|\nabla_x \log(\rho(t, x))\|_2^2 \right) \rho(t, x) \, dx \, dt, \quad (4)$$

where the infimum is taken over time-dependent probability measures $\rho(t, x)$ that interpolate between $\mu$ at $t = 0$ and $\nu$ at $t = 1$, and time-dependent vector fields $v(t, x)$ under the continuity equation constraint $\partial_t \rho(t, x) + \text{div}(\rho(t, x)v(t, x)) = 0$ where div is the usual divergence operator. The first term in the r.h.s. of Eq. (4) is the kinetic energy and the second is the Fisher information integrated in time. For $\lambda \geq 0$, there exists a unique minimizer of the r.h.s. [30] denoted by $\rho_\lambda$ and we define

$$I_\lambda(\mu, \nu) \overset{\text{def.}}{=} \int_0^1 \int_{\mathbb{R}^d} \|\nabla_x \log(\rho_\lambda(t, x))\|_2^2 \, \rho_\lambda(t, x) \, dx \, dt. \quad (5)$$

Remark that $I_0(\mu, \mu)$ is the Fisher information of $\mu$ and $I_0(\mu, \nu)$ is the Fisher information of the Wasserstein geodesic between $\mu$ and $\nu$. Building on [14], we next show that the Sinkhorn divergence approximates $W_2^2(\mu, \nu)$ with an error in $O(\lambda^2)$, as suggested by Eq. (4).

**Theorem 1.** *Assume that $\mu, \nu \in \mathcal{P}(\mathbb{R}^d)$ have bounded densities and supports. It holds*

$$\left| S_\lambda(\mu, \nu) - W_2^2(\mu, \nu) \right| \leq \frac{\lambda^2}{4} \max \left\{ I_0(\mu, \nu), (I_0(\mu, \mu) + I_0(\nu, \nu))/2 \right\}.$$

*If moreover $I_0(\mu, \nu), I_0(\mu, \mu), I_0(\nu, \nu) < \infty$ then*

$$S_\lambda(\mu, \nu) - W_2^2(\mu, \nu) = \frac{\lambda^2}{4} \left( I_0(\mu, \nu) - (I_0(\mu, \mu) + I_0(\nu, \nu))/2 \right) + o(\lambda^2).$$

*Proof.* Denote the right-hand side of (4) by $J_{\lambda^2}(\mu, \nu)$ and note that $S_\lambda(\mu, \nu) - W_2^2(\mu, \nu) = (J_{\lambda^2}(\mu, \nu) - J_0(\mu, \nu)) - (J_{\lambda^2}(\mu, \mu) + J_{\lambda^2}(\nu, \nu))/2$ and $J_0(\mu, \mu) = J_0(\nu, \nu) = 0$. Since $\rho_0$ is feasible in Eq. (4), we have $J_0(\mu, \nu) \leq J_{\lambda^2}(\mu, \nu) \leq J_0(\mu, \nu) + (\lambda^2/4)I_0(\mu, \nu)$, hence the bound. For the second claim, we prove in Appendix A (Lemma 1) that the right derivative at 0 of $\sigma \mapsto J_\sigma$ is $\frac{1}{4}I_0(\mu, \nu)$, which justifies the Taylor expansion. $\square$

The Fisher information of $\mu$ or $\nu$ can be bounded by assuming regularity of the densities, but bounding $I_0(\mu, \nu)$ is more subtle. Next, we bound $I_0(\mu, \nu)$ assuming regularity on the *Brenier potential* $\varphi$, which is the convex function such that $\nabla \varphi$ is the optimal transport map from $\mu$ to $\nu$ [52].

**Proposition 1.** *Let $\mu, \nu \in \mathcal{P}(\mathbb{R}^d)$ be absolutely continuous with compact support. Assume that the Brenier potential $\varphi$ has a Hessian satisfying $0 \prec \kappa \mathrm{Id} \preceq \nabla^2 \varphi \preceq K\mathrm{Id}$ and that $\nabla^2 \varphi$ is L-Lipschitz continuous, then $I_0(\mu, \nu) \leq 2\kappa^{-1}(I_0(\mu, \mu) + \kappa^{-2}L^2/3)$. In particular, if $\varphi$ is quadratic then $I_0(\mu, \nu) \leq 2\kappa^{-1}I_0(\mu, \mu)$. If $d = 1$, then $I_0(\mu, \nu) \leq \frac{2}{3}(\kappa^{-1}I_0(\mu, \mu) + KI_0(\nu, \nu))$.*

Sufficient conditions on the densities of $\mu$ and $\nu$ to guarantee bounds on $\nabla^2 \varphi$ are known (e.g. bounds on their first derivative and on their log-densities over their convex support [17, Thm 3.3]). However, the assumption that $\nabla^2 \varphi$ is Lipschitz continuous is more demanding and potentially not sharp as it can be avoided when $d = 1$. Note that the Brenier potential $\varphi$ is quadratic whenever the densities are in the same family of elliptically contoured distributions [6]. For Gaussian densities, we show in Appendix A that $I_0(\mu, \nu)$ admits an explicit expression, given in Section 4.

## 3 Performance analysis of the Sinkhorn divergence estimator

In this section, we discuss the performance of the Sinkhorn divergence estimator in two situations: when we observe independent samples or when we have access to discretized densities. But first, we study the plug-in estimator, which is the baseline against which our estimators are compared.

### 3.1 Analysis of the plug-in estimator

**A tighter statistical bound for the plug-in estimator.** Let us first study the rate of convergence of $W_2^2(\hat{\mu}_n, \hat{\nu}_n)$ towards $W_2^2(\mu, \nu)$ where $\hat{\mu}_n$ and $\hat{\nu}_n$ are empirical distributions of $n$ independent samples. This is well-studied in the case $\mu = \nu$, but the case $\mu \neq \nu$ was not specifically covered in the literature except for discrete measures [55].

**Theorem 2.** *If $\mu, \nu \in \mathcal{P}(\mathbb{R}^d)$ are supported on a set of diameter 1 then it holds*

$$\mathbf{E}\big[|W_2^2(\hat{\mu}_n, \hat{\nu}_n) - W_2^2(\mu, \nu)|\big] \lesssim \begin{cases} n^{-2/d} & \text{if } d > 4, \\ n^{-1/2}\log(n) & \text{if } d = 4, \\ n^{-1/2} & \text{if } d < 4, \end{cases}$$

*where the notation $\lesssim$ hides constants that only depend on the dimension $d$. Also, this estimator concentrates well around its expectation, in the sense that for all $t \geq 0$,*

$$\mathbf{P}\Big[|W_2^2(\hat{\mu}_n, \hat{\nu}_n) - \mathbf{E}[W_2^2(\hat{\mu}_n, \hat{\nu}_n)]| \geq t\Big] \leq 2\exp(-nt^2).$$

To prove this result in Appendix C, we first upper bound the expected error by the Rademacher complexity of a certain set of convex and Lipschitz functions. We use Dudley's chaining and a bound on the covering number of this set of functions due to Bronshtein [10] to conclude. The concentration bound is already present in a similar form in [59, Prop. 20]. When $\mu = \nu$, this bound is well-known and has a sharp exponent [59, 54, 8, 19, 24]. However, perhaps surprisingly, this result implies that the plug-in estimator $W_2(\hat{\mu}_n, \hat{\nu}_n)$ (without the square) converges at the rate $n^{-2/d}$ when $\mu \neq \nu$, while only a bound in $n^{-1/d}$ (the rate when $\mu = \nu$) was known. This is the content of the following corollary. See Figure 1 for a numerical illustration of these rates.

**Corollary 1.** *Assume that $\mu, \nu$ are supported on a set of diameter 1 and satisfy $W_2(\mu, \nu) \geq \alpha > 0$. Then $\mathbf{E}\big[|W_2(\mu_n, \nu_n) - W_2(\mu, \nu)|\big]$ enjoys the bound given in Theorem 2 multiplied by $1/\alpha$.*

*Proof.* It is sufficient to take expectations in the following inequality :

$$|W_2(\hat{\mu}_n, \hat{\nu}_n) - W_2(\mu, \nu)| = \frac{|W_2^2(\hat{\mu}_n, \hat{\nu}_n) - W_2^2(\mu, \nu)|}{W_2(\mu, \nu) + W_2(\hat{\mu}_n, \hat{\nu}_n)} \leq \frac{1}{\alpha}|W_2^2(\hat{\mu}_n, \hat{\nu}_n) - W_2^2(\mu, \nu)|. \quad \square$$

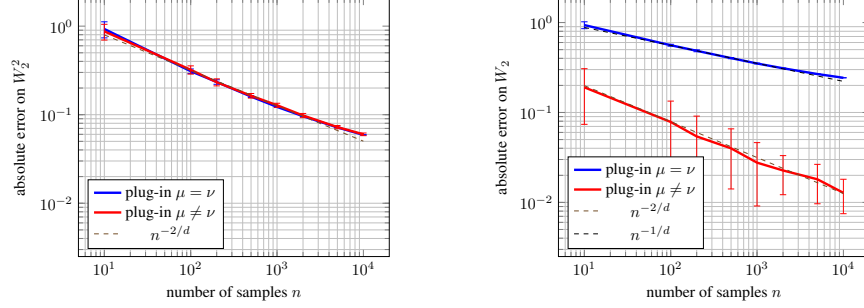

Figure 1: Estimation error of the plug-in estimator for $\mu, \nu$ compactly supported with $d = 5$ (as detailed in Appendix G). Left: error on the cost $W_2^2$ has rate $n^{-2/d}$ (Theorem 2). Right: error on $W_2$ has rate $n^{-1/d}$ if $\mu = \nu$ and $n^{-2/d}$ if $\mu \neq \nu$ (Corollary 1) with $\mathbf{E}_\mu[x] = 0$ and $\mathbf{E}_\nu[x] = (1, \dots, 1)$.

**Computational complexity via Sinkhorn's algorithm.** In previous work [2, 20], solving $T_\lambda(\hat{\mu}_n, \hat{\nu}_n)$ with $\lambda > 0$ has been studied as a computationally efficient way to compute $T_0(\hat{\mu}_n, \hat{\nu}_n)$ and related quantities. One standard algorithm to compute $T_\lambda$ is Sinkhorn's algorithm, which can be interpreted as alternate block maximization on the dual of Eq. (1), see Appendix B. Given two discrete marginals $\hat{\mu}_n = \sum_{i=1}^n p_i \delta_{x_i}$ and $\hat{\nu}_n = \sum_{i=1}^n q_j \delta_{y_j}$, let us define the cost matrix with entries $c_{i,j} = \frac{1}{2}\|x_i - y_j\|_2^2$. The iterates $u^{(k)}, v^{(k)} \in \mathbb{R}^n$, $k \geq 1$ of Sinkhorn's algorithm are defined as follows: let $v^{(0)} = 0 \in \mathbb{R}^n$ and let

$$u_i^{(k)} = -\lambda \log \Big( \sum_{j=1}^n e^{(v_j^{(k-1)} - c_{i,j})/\lambda} q_j \Big) \quad \text{and} \quad v_j^{(k)} = -\lambda \log \Big( \sum_{i=1}^n e^{(u_i^{(k)} - c_{i,j})/\lambda} p_i \Big). \quad (6)$$

An estimate for $\hat{T}_{\lambda,n} \overset{\text{def.}}{=} T_\lambda(\hat{\mu}_n, \hat{\nu}_n)$ is then given by $\hat{T}_{\lambda,n}^{(k)} = 2 \sum_{i=1}^n u_i^{(k)} p_i + v_i^{(k)} q_i$. These iterations enjoy the following guarantee, proved in [20] (see details in Appendix B).

**Proposition 2.** *It holds $|\hat{T}_{\lambda,n}^{(k)} - \hat{T}_{\lambda,n}| \leq 2\|c\|_\infty^2/(\lambda k)$ where $\|c\|_\infty = \max_{i,j} \|x_i - y_j\|_2^2/2$.*

In particular, taking into account the fact that each iteration requires $O(n^2)$ arithmetic operations, Sinkhorn's algorithm returns an $\varepsilon$-accurate estimation of $\hat{T}_{\lambda,n}$ in time $O(n^2 \|c\|_\infty^2/(\lambda \varepsilon))$. Moreover, if $\alpha > 0$ is such that $p_i, q_j \geq \alpha/n$, we have the approximation bound $|\hat{T}_{\lambda,n} - \hat{T}_{0,n}| \leq 4\lambda \log(n/\alpha)$ which follows by bounding the relative entropy of admissible transport plans [2]. By fixing $\lambda = \varepsilon/4(\log(n/\alpha))$, we thus obtain an $\varepsilon$-accurate estimation of $\hat{T}_{0,n}$ in $O(n^2 \log(n/\alpha)\|c\|_\infty^2/\varepsilon^2)$ operations. As a consequence, by combining Theorem 2 and Proposition 2, we can thus give the following computational complexity bound to estimate $W_2^2(\mu, \nu)$ given random samples that takes into account the number of samples and the regularization level required to reach a certain accuracy.

**Proposition 3.** *Assume that $\mu, \nu$ are supported on a set of diameter 1. Using $\hat{T}_{\lambda,n}^{(k)}$, an $\varepsilon$-accurate estimation of $W_2^2(\mu, \nu)$ is achieved with probability $1 - \delta$ in $\tilde{O}(\varepsilon^{-\max\{6, d+2\}})$ operations, where $\tilde{O}$ hides poly-log factors in $1/\varepsilon$ and $1/\delta$.*

*Proof idea.* We write $W_2^2 \overset{\text{def.}}{=} W_2^2(\mu, \nu)$, $\hat{W}_2^2 \overset{\text{def.}}{=} W_2^2(\hat{\mu}_n, \hat{\nu}_n)$ and consider the error decomposition

$$|\hat{T}_{\lambda,n}^{(k)} - W_2^2| \leq |\hat{T}_{\lambda,n}^{(k)} - \hat{T}_{\lambda,n}| + |\hat{T}_{\lambda,n} - \hat{W}_2^2| + |\hat{W}_2^2 - \mathbf{E}[\hat{W}_2^2]| + \mathbf{E}|\hat{W}_2^2 - W_2^2]|$$

where each term has been bounded in the previous discussion, see details in Appendix C. □

### 3.2 Performance of the Sinkhorn divergence estimator given random samples

**Statistical performance.** Let us now turn to our object of interest which is the Sinkhorn divergence estimator $\hat{S}_{\lambda,n} \overset{\text{def.}}{=} S_\lambda(\hat{\mu}_n, \hat{\nu}_n)$, defined from $n$ independent samples from $\mu$ and $\nu$. We note that all the results in this section also apply to the estimator $T_\lambda(\hat{\mu}_n, \hat{\nu}_n) - (T_\lambda(\hat{\mu}_{n/2}, \hat{\mu}'_{n/2}) + T_\lambda(\hat{\nu}_{n/2}, \hat{\nu}'_{n/2}))/2$ where $\hat{\mu}_{n/2}$ (resp. $\hat{\mu}'_{n/2}$) is the empirical distribution of the first (resp. second) half samples from $\mu$ (assuming $n$ even for conciseness), which is a natural alternative definition. The following result gives the expected error of the estimator $\hat{S}_{\lambda,n}$.

**Proposition 4.** *Let $\mu, \nu$ be supported on a set of diameter 1 and assume that $|S_\lambda(\mu, \nu) - W_2^2(\mu, \nu)| \le \lambda^2 I$ for some $I > 0$ (see guarantees in Section 2). Then, with the choice $\lambda = n^{\frac{-1}{d'+4}}$, it holds*

$$\mathbf{E}\big[|\hat{S}_{\lambda,n} - W_2^2(\mu, \nu)|\big] \lesssim n^{\frac{-2}{d'+4}}.$$

*where $d' = 2\lfloor d/2 \rfloor$ and $\lesssim$ hides a constant depending only on $I$ and $d$. Also, this estimator concentrates well around its expectation: for all $t, \lambda \ge 0$, $\mathbf{P}\Big[|\hat{S}_{\lambda,n} - \mathbf{E}[\hat{S}_{\lambda,n}]| \ge t\Big] \le 2\exp(-nt^2/4)$.*

Observe that when $d$ is large, the exponent $-2/(d'+4)$ is equivalent to $-2/d$ which is the rate of the plug-in estimator as shown in Theorem 2. However, except for $d = 1$, this exponent is slightly worse and we believe that this is due to a weakness in our bound. In fact, in our numerical experiments we observe that $\hat{S}_{\lambda,n}$ is in fact more statistically efficient than the plug-in estimator (cf. Figure 2).

**Computational performance.** An ideal theoretical goal would be to exhibit a computational advantage for using $\hat{S}_{\lambda,n}$ in the sense of Proposition 3, but unfortunately the statistical bound in Proposition 4 is not strong enough to allow for such a result. Still, there is a clear computational advantage in using $\hat{S}_{\lambda,n}$ which is that to attain an accuracy $\varepsilon$, it requires a regularization level $\lambda$ of order $\varepsilon^{1/2}$ instead of $\varepsilon$ for the plug-in estimator. This advantage can be formalized as follows, where $\hat{S}_{\lambda,n}^{(k)}$ is the estimation of $\hat{S}_{\lambda,n}$ obtained after $k$ Sinkhorn's iterations.

**Proposition 5.** *Under the assumptions of Proposition 4, an $\varepsilon$-accurate estimation of $W_2^2(\mu, \nu)$ can be obtained with probability $1 - \delta$ in $\tilde{O}(\varepsilon^{-(d'+5.5)})$ computations via $\hat{S}_{\lambda,n}^{(k)}$ where $d' = 2\lfloor d/2 \rfloor$ and $\tilde{O}$ hides a poly-log factor in $1/\delta$. Given $n$ samples, both estimators can achieve with probability $1 - \delta$ an accuracy $\varepsilon \asymp n^{-2/(d'+4)}$, but in time $\tilde{O}(n^2 \varepsilon^{-1.5})$ via $\hat{S}_{\lambda,n}^{(k)}$ and in time $\tilde{O}(n^2 \varepsilon^{-2})$ via $T_{\lambda,n}^{(k)}$.*

*Proof idea.* For $\hat{T}_{\lambda,n}^{(k)}$, we consider the error decomposition of Proposition 3, while for $\hat{S}_{\lambda,n}^{(k)}$, we write

$$|\hat{S}_{\lambda,n}^{(k)} - W_2^2| \le |\hat{S}_{\lambda,n}^{(k)} - \hat{S}_{\lambda,n}| + |\hat{S}_{\lambda,n} - \mathbf{E}[\hat{S}_{\lambda,n}]| + \mathbf{E}|\hat{S}_{\lambda,n} - S_\lambda| + |S_\lambda - W_2^2|.$$

The key difference with the decomposition in the proof of Proposition 3 is that the error induced by the entropic regularization is bounded on the population quantities instead of the empirical ones. These terms have been bounded in the previous discussion, see details in Appendix D. $\qquad\square$

### 3.3 Performance of the Sinkhorn divergence estimator given densities discretized on grids

In this section, we consider the case where the marginals $\mu$ and $\nu$ are not randomly sampled, but instead are accessed via their discretized densities which is the common situation in imaging sciences. We show a stability property of the entropy regularized optimal transport which leads to improved error bounds compared to the plug-in estimator.

For simplicity, we consider measures on the $d$ dimensional torus $\mathbb{T}^d = (\mathbb{R}/\mathbb{Z})^d$ with its usual distance denoted by $\|[x-y]\|_2$. For a measure $\mu \in \mathcal{P}(\mathbb{T}^d)$ its discretization $\mu_h$ at resolution $h = 1/m$ for an integer $m$ is the discrete measure with $n = m^d$ atoms supported on the regular grid $(\mathbb{Z}/m\mathbb{Z})^d$ which gives to each point the mass of $\mu$ on its surrounding cell. The following approximation result suggests that regularizing the optimal transport problem increases the stability under such a discretization.

**Proposition 6** (Stability under discretization)**.** *Assume that $\mu, \nu \in \mathcal{P}(\mathbb{T}^d)$ admit $M$-Lipschitz continuous log-densities and let $C > 0$ be any constant. If $h(M + \lambda^{-1}) \le C$ then*

$$|T_\lambda(\mu_h, \nu_h) - T_\lambda(\mu, \nu)| \lesssim \min\{h, h^2(\lambda^{-1} + M + 1)\}$$

*where $\lesssim$ hides constants that only depend on $d$ and $C$.*

This bound implies an error of order $h^2$ for the entropy regularized problem while it is not known whether such a bound is possible for $\lambda = 0$, where a naive analysis suggests a bound of order $h$. When combined with the approximation error and the analysis of Sinkhorn's iterations, this yields the following performance guarantees for $S_\lambda(\mu_h, \nu_h)$ as defined in Eq. (2).

**Proposition 7.** *Assume that $\mu, \nu \in \mathcal{P}(\mathbb{T}^d)$ admit Lipschitz continuous log-densities and that $I_0(\mu, \nu)$ is finite. We can estimate $W_2^2(\mu, \nu)$ to $\varepsilon$-accuracy:*

– with $T_\lambda(\mu_h, \nu_h)$ in time $\tilde{O}(\varepsilon^{-(2d+2)})$ by setting $h \asymp \varepsilon$ and $\lambda \asymp \varepsilon/\log(1/\varepsilon)$,

– with $S_\lambda(\mu_h, \nu_h)$ in time $O(\varepsilon^{-(3d/2+3/2)})$ by setting $h \asymp \varepsilon^{3/4}$ and $\lambda \asymp \varepsilon^{1/2}$.

This result suggests that $S_\lambda(\mu_h, \nu_h)$ estimates $W_2^2(\mu, \nu)$ both faster and more accurately than $T_\lambda(\mu_h, \nu_h)$ for their respective optimal $\lambda$, and this behavior is observed in numerical experiments (cf. Figure 4). Our aim with Proposition 7 is to illustrate the potential usefulness of the debiasing terms beyond the random sampling setting, but we stress that we are just comparing simple upper bounds which are not intended to be the best possible (in particular, we are not exploiting the fact that the computational cost of each Sinkhorn iteration could be reduced from $O(n^2)$ to $O(n \log(n))$ using discrete convolutions [5, Sec. 6.3.1]). In fact, in a similar setting, a completely different analysis of Sinkhorn's iterations is carried in [5, Cor.1.4], where a time complexity in $\tilde{O}(\varepsilon^{-(2d+1)})$ is derived for $T_\lambda(\mu_h, \nu_h)$.

## 4  Towards faster estimation with Richardson extrapolation

The systematic bias induced by the Fisher information terms in Theorem 1 can be removed using Richardson extrapolation [35, 51], which usefulness in machine learning was recently pointed out in [4]. This technique consists in taking linear combinations of $S_\lambda$ for various values of $\lambda > 0$ in order to estimate $S_0$, by cancelling the successive terms of the Taylor expansion of $S_\lambda$ at $0$. Since in our context the first term of $S_\lambda - S_0$ is of order $\lambda^2$, this suggests to define (among other possible choices) $R_\lambda \stackrel{\text{def.}}{=} 2S_\lambda - S_{\sqrt{2}\lambda}$. Indeed, whenever $S_\lambda = S_0 + \lambda^2 I + o(\lambda^2)$ for some $I \in \mathbb{R}$, such as under the assumptions of Theorem 1, this quantity satisfies $R_\lambda = S_0 + o(\lambda^2)$.

**Efficiency of $R_\lambda$ under an abstract assumption.**  A difficulty with $R_\lambda$, or other extrapolated estimators, is that understanding their performance requires a fine understanding of the regularization path $\lambda \mapsto S_\lambda$. By remarking that in Eq. (4), $\lambda$ appears only via its square after debiasing, we might conjecture that if $S_\lambda$ admits a 4th order Taylor expansion at $\lambda = 0$, then the third term vanish. Before giving some arguments in favor of this property, let us state what it implies in terms of the performance of $\hat{R}_{\lambda,n} = \hat{S}_{\lambda,n} - \hat{S}_{\sqrt{2}\lambda,n}$, the extrapolation of the estimator $\hat{S}_{\lambda,n}$.

**Proposition 8.**  *Assume that $\mu, \nu$ are compactly supported, that $S_\lambda(\mu, \nu) - W_2^2(\mu, \nu) = \lambda^2 I + O(\lambda^4)$ for some $I \in \mathbb{R}$ and let $d' = 2\lfloor d/2 \rfloor$. Then with $\lambda \asymp n^{-1/(d'+8)}$ it holds*

$$\mathbf{E}\big[|\hat{R}_{\lambda,n} - W_2^2(\mu, \nu)|\big] \lesssim n^{-4/(d'+8)}.$$

*Moreover, with probability $1 - \delta$, this estimator returns an $\varepsilon$-accurate estimation of $W_2^2(\mu, \nu)$ with $\tilde{O}(\varepsilon^{-(d'+11)/2})$ computations via Sinkhorn's algorithm where $\tilde{O}$ hides poly-log factors in $1/\delta$.*

*Proof.*  We use Lemma 5 to get

$$\mathbf{E}[|\hat{R}_{\lambda,n} - W_2^2(\mu, \nu)|] \leq \mathbf{E}[|\hat{R}_{\lambda,n} - R_\lambda(\mu, \nu)|] + |R_\lambda(\mu, \nu) - W_2^2(\mu, \nu)| \lesssim (1 + \lambda^{-d'/2})n^{-1/2} + \lambda^4$$

and optimize the bound in $\lambda$. For the last claim we proceed as in the proof of Proposition 3.  □

Under this abstract assumption, there is thus a clear statistical improvement over the plug-in estimator for $d > 8$ and a computational improvement for $d > 6$. Notice that a similar performance analysis could be done in the deterministic setting of Section 3.3. In the rest of this section we discuss the assumption of Proposition 8. First we show that it is satisfied in the Gaussian case and second we propose formal calculations towards a 4th order Taylor expansion of $T_\lambda$.

**Gaussian case.**  Let $\mu = \mathcal{N}(a, A)$ and $\nu = \mathcal{N}(b, B)$ be Gaussian probability distributions with means $a, b \in \mathbb{R}^d$ and positive definite covariances $A, B \in \mathbb{R}^{d \times d}$. In this case, it is well known that $W_2^2(\mu, \nu) = \|a - b\|_2^2 + \mathrm{d}_\mathrm{b}^2(A, B)$ where $\mathrm{d}_\mathrm{b}^2(A, B) \stackrel{\text{def.}}{=} \mathrm{tr}(A) + \mathrm{tr}(B) - 2\,\mathrm{tr}(S)$ with $S = (A^{1/2}BA^{1/2})^{1/2}$ is the squared Bures distance [6]. More recently, an explicit expression for $T_\lambda(\mu, \nu)$ was derived in [34, 11, 38]. By a Taylor expansion of this expression (see Appendix F), we find that

$$S_\lambda(\mu, \nu) - W_2^2(\mu, \nu) = -\frac{\lambda^2}{8}\mathrm{d}_\mathrm{b}^2(A^{-1}, B^{-1}) + \frac{\lambda^4}{384}\mathrm{d}_\mathrm{b}^2(A^{-3}, B^{-3}) + O(\lambda^5).$$

This expansion shows that the hypotheses of Proposition 8 are satisfied (to the exception of the compactness assumption, but note that sample complexity bounds for $S_\lambda$ are also known in this case [40]). Also we can explicitly compute the Fisher information $I_0(\mu, \nu) = \mathrm{tr}(S^{-1})$ (Appendix A) which shows that the second order term is consistent, as it must, with the expansion in Theorem 1.

**Formal fourth order expansion.** Denoting $J_{\lambda^2}(\mu, \nu)$ the r.h.s. of Eq. (4), we show in Lemma 1 that $\sigma \mapsto J_\sigma$ admits a right derivative at all $\sigma \geq 0$ which is the Fisher information $\frac{1}{4} I_{\sqrt{\sigma}}(\mu, \nu)$ defined in Eq. (5). Thus, if we assume that $\sigma \mapsto I_{\sqrt{\sigma}}(\mu, \nu)$ admits a right derivative $I_0'$ at 0, then it holds

$$T_\lambda(\mu, \nu) = T_0(\mu, \nu) - d\lambda \log(2\pi\lambda) - \lambda(H(\mu) + H(\nu)) + \frac{\lambda^2}{4} I_0(\mu, \nu) + \frac{\lambda^4}{8} I_0' + o(\lambda^4),$$

where $I_0' = \frac{\mathrm{d}}{\mathrm{d}(\lambda^2)} I_\lambda(\mu, \nu)|_{\lambda=0} = \int_0^1 \int_{\mathbb{R}^d} (\|\nabla \log \rho_0\|^2 - 2\Delta\rho_0/\rho_0)\delta_{\lambda^2}\rho_\lambda|_{\lambda=0} \, dx$ is the variation of Fisher information in the direction of $\delta_{\lambda^2}\rho_\lambda|_{\lambda=0_+}$, the first variation of $\rho_\lambda$ w.r.t. $\lambda^2$. Hence under this abstract regularity assumption on $I_{\sqrt{\sigma}}(\mu, \nu)$, the result of Proposition 8 holds true.

# 5   Numerical experiments

In this section, we assess the statistical and computational efficiency of the proposed estimators on synthetic problems[2]. While this is what our theory controls, the error on the scalar $W_2^2(\mu, \nu)$ is not a suitable quantity to plot as it might vanish spuriously as we vary other parameters (such as $n$ or $\lambda$), which hinders interpretation of the plots (see Appendix G). Instead, we propose to observe a more stringent and stable quantity, namely the $L_1$ error on the estimated dual potential $\varphi$, which is the Lagrange multiplier associated to the first marginal constraint in Eq. (1). This dual potential is the gradient of $W_2^2(\mu, \nu)$ with respect to $\mu$ [52, Prop. 7.17], a quantity of high interest when training machine learning models with $W_2^2$ as a loss function.

Specifically, given $v^{(k)} \in \mathbb{R}^n$ obtained after $k$ Sinkhorn's iterations with discrete marginals $\mu_n, \nu_n$ as in Eq. (6), we define the function $\hat{u}_{\mu,\nu}(x) = -\lambda \log(\sum_{j=1}^n e^{(v_j^{(k)} - \frac{1}{2}\|x-y_j\|_2^2)/\lambda} q_j)$. The quantity we plot is $\int |\hat{\varphi}_{\lambda,n}(x) - \varphi(x)| \mathrm{d}\mu(x)$ estimated via Monte Carlo integration or on a fine grid, where $\hat{\varphi}_{\lambda,n}$ is defined as follows: (i) $\hat{\varphi}_{\lambda,n} = 2\hat{u}_{\mu,\nu}$ for the biased estimator $\hat{T}_{\lambda,n}$, (ii) $\hat{\varphi}_{\lambda,n} = 2\hat{u}_{\mu,\nu} - (\hat{u}_{\mu,\mu'} + \hat{v}_{\mu,\mu'})$ for the debiased estimator $\hat{S}_{\lambda,n}$ and (iii) $2\hat{\varphi}_{\lambda,n} - \hat{\varphi}_{\sqrt{2}\lambda,n}$ for the extrapolated estimator $\hat{R}_{\lambda,n}$.

**Random sampling.** Figure 2 shows the approximation error for the estimators $T_\lambda$, $S_\lambda$ and $R_\lambda$ in the random sampling setting. Here, $\mu, \nu \in \mathcal{P}(\mathbb{R}^d)$ with $d = 5$ are smooth elliptically contoured distributions with compact support and are such that the optimal potential $\varphi$ is quadratic and admits a closed-form, as well as the transport cost (see Appendix G). These properties guarantee that the conclusions of Proposition 4 apply. As expected, for a given $\lambda$, $S_\lambda$ and $R_\lambda$ have a much smaller bias than $T_\lambda$ (left plot). Looking at the performance as a function of $\lambda$ (middle plot), we see that the error is minimal for some $\lambda^*$ that is much larger than what is needed for $T_\lambda$ to achieve a comparable accuracy. Also, choosing the best $\lambda^*$ for each $n$ (right panel), we see that $S_{\lambda^*}$ has the same rate as the plug-in estimator (estimated with $T_\lambda$ with a small $\lambda$), with a better constant. We remark that $R_\lambda$ does not converge faster, which does not contradict ours results since we have no guarantee on the specific quantity plotted here.

Overall, these estimators require less samples and a larger $\lambda$ to achieve a given accuracy compared to $T_\lambda$, which leads to substantial computational gains. This is illustrated on Figure 3 where for a target $L^1$ error on the potential, we chose the largest $\lambda$ and smallest $n$ that achieve this error, with $\lambda \in [0.1, 1]$ and $n \in [10, 100000]$. We report the computational time using the Sinkhorn's iterations of Eq. (6) stopped when the $\ell_1$-error on the marginals is below $10^{-5}$. We observe that for small target accuracies, the estimators $S_\lambda$ and $R_\lambda$ compare favorably to $T_\lambda$. In practical settings, one does not know *a priori* the best choice for $\lambda$, but many machine learning tasks involving $W_2^2$ come with a performance criterion, in which case cross-validation can be used to select this parameter.

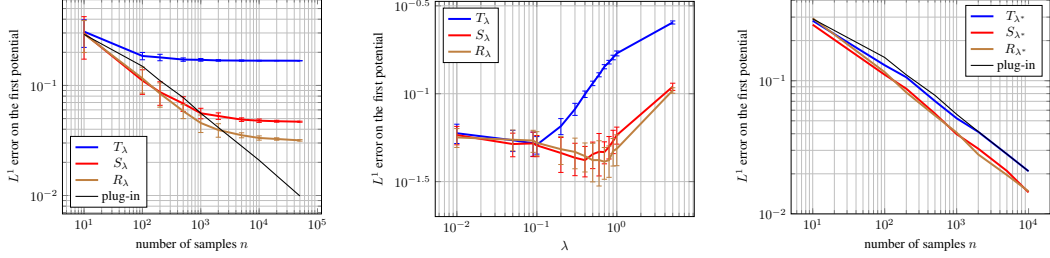

Figure 2: $L^1$ estimation error on the first potential for $\mu, \nu$ smooth compactly supported distributions with $d = 5$. Left: as function of $n$ for $\lambda = 1$. Middle: as a function of $\lambda$, for $n = 10000$. Right: as a function of $n$ for the optimal $\lambda^*(n)$. Error bars show the standard deviation on 30 realizations.

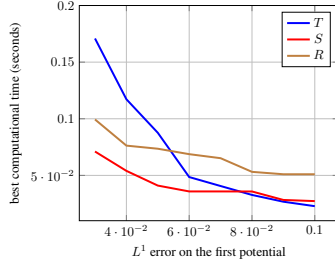

Figure 3: Best computational time achieved by the estimators to reach a given accuracy (after optimizing over $n$ and $\lambda$), for $\mu, \nu$ smooth compactly supported distributions with $d = 5$.

**Discretization on grids.** Figure 4 shows the evolution of the errors for densities $(\mu, \nu)$ on the 1-D torus, the setting of Proposition 7. In this case, one can compute efficiently the dual potentials $\varphi$ using cumulative functions [48]. This figure shows that, as expected, for a fixed $(h, \lambda)$ the error of $S_\lambda$ and $R_\lambda$ is systematically lower than that of $T_\lambda$. Even when selecting the optimal regularization $\lambda^\star(h)$ for each $h$ and for each method (which is a fair comparison), the error of $S_\lambda$ and $R_\lambda$ is still lower. Furthermore, the optimal parameter $\lambda^\star(h)$ is systematically larger for $S_\lambda$ and $R_\lambda$. Additional figures showing visual comparisons of the potentials and their approximations are provided in the appendix.

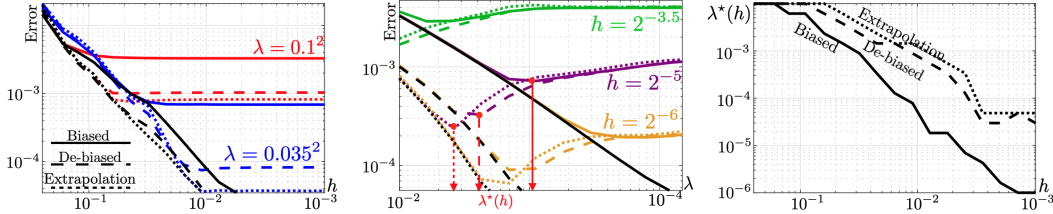

Figure 4: Left: $L^1$ error on the first potential $\varphi$ as a function of the grid size $h$, for several value of $\lambda$. Middle: same error, displayed as a function of $\lambda$, for several grid sizes $h$. Right: evolution of the optimal regularization parameter $\lambda^\star(h)$ as a function of the grid size $h$.

# 6 Conclusion and open questions

In this paper we have exhibited the usefulness of entropic regularization with debiasing for the estimation of the squared Wasserstein distance: it may increase both accuracy and efficiency when the problem has a smooth nature. Numerical experiments suggest that the theory could be extended in several directions. First, the Sinkhorn divergence estimator appears at least as statistical efficient as the plug-in estimator, while our bound is slightly weaker. Also, the estimation of Kantorovich potentials seems to enjoy similar guaranties, but this is not covered by our theory.

## Broader Impact

Broader impact statement does not apply for this paper, which is of theoretical nature.

## Acknowledgments

The works of Pierre Roussillon, Flavien Léger and Gabriel Peyré is supported by the ERC grant NORIA and by the French government under management of Agence Nationale de la Recherche as part of the "Investissements d'avenir" program, reference ANR19-P3IA-0001 (PRAIRIE 3IA Institute).

## Footnotes

[2]The code to reproduce these experiments is available at this webpage `https://gitlab.com/proussillon/wasserstein-estimation-sinkhorn-divergence`.

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
