[Supplementary Material]

## Supplementary Material

Supplementary material for the paper: "Faster Wasserstein Distance Estimation with the Sinkhorn Divergence" authored by Lénaïc Chizat, Pierre Roussillon, Flavien Léger, François-Xavier Vialard and Gabriel Peyré (NeurIPS 2020). This supplementary material is organized as follows:

- Appendix A contains the proofs of Section 2,
- Appendix B recaps the convergence analysis of [20] to obtain Proposition 2,
- Appendix C contains the proofs of Section 3.1,
- Appendix D contains the proofs of Section 3.2,
- Appendix E contains the proofs of Section 3.3,
- in Appendix F, we derive the Taylor expansion for Gaussian distributions presented in Section 4,
- finally, Appendix G contains details on the settings of the numerical experiments and additional figures.

## A    Bounds on the approximation error

**Dynamic entropy regularized optimal transport.**    Let us first justify how to obtain Eq. (4) since our conventions are slightly different than in [14]. In that reference, for $\mu$ and $\nu$ absolutely continuous with compact support, the authors define

$$\lambda C_\lambda(\mu, \nu) = \min_{\gamma \in \Pi(\mu,\nu)} \lambda H(\gamma, K)$$

where $K = (2\pi\lambda)^{-d/2} \exp(-\|y - x\|_2^2/(2\lambda))\mathrm{d}x\mathrm{d}y$ is the heat kernel at time $\lambda/2$. In contrast, we can see from Eq. (1) that

$$\frac{1}{2}T_\lambda(\mu, \nu) = \min_{\gamma \in \Pi(\mu,\nu)} \lambda H(\gamma, \tilde{K})$$

where $\tilde{K} = \exp(-\|y - x\|_2^2/(2\lambda))\mu(x)\nu(y)\mathrm{d}x\mathrm{d}y$.    We directly deduce that $\frac{1}{2}T_\lambda(\mu, \nu) = \lambda C_\lambda(\mu, \nu) - \lambda H(\mu) - \lambda H(\nu) - \frac{d\lambda}{2}\log(2\pi\lambda)$. Thus Eq. (4) follows by the dynamic formulation of entropy regularized optimal transport in [14] which reads

$$\lambda C_\lambda(\mu, \nu) - \frac{\lambda}{2}H(\mu) - \frac{\lambda}{2}H(\nu) = \min_{\rho, v} \int_0^1 \int_{\mathbb{R}^d} \left(\frac{1}{2}\|v(t,x)\|_2^2 + \frac{\lambda^2}{8}\|\nabla_x \log \rho(t,x)\|_2^2\right) \rho(t,x)\mathrm{d}x\mathrm{d}t$$

where the constraints on $(\rho, v)$ are as in Eq. (4). Note that $\nabla_x \log \rho$ refers to the *weak logarithmic gradient* of $\rho$, which in particular does not requires $\rho > 0$ to be well defined, but only that for almost every $t \in [0,1]$, $\rho(t, \cdot)$ admits a distributional gradient which is an absolutely continuous measure with respect to $\rho(t, \cdot)$, and $\nabla_x \log \rho_t \stackrel{\text{def.}}{=} \frac{\mathrm{d}\nabla \rho_t}{\mathrm{d}\rho_t}$ refers to its density with respect to $\rho_t$ (see e.g. [29]).

**First order expansion.**    Let us state and prove a lemma that intervenes in the proof of Theorem 1. Arguments towards this expansion appeared in [14, Theorem 1.6] but under an abstract twice-differentiability assumption that is not needed in our statement.

**Lemma 1.** *Assume that $\mu, \nu \in \mathcal{P}(\mathbb{R}^d)$ have bounded densities and supports. It holds*

$$\frac{\mathrm{d}}{\mathrm{d}\sigma}J_\sigma|_{\sigma=0_+} = \frac{1}{4}I_0(\mu, \nu)$$

*where, as in the proof of Theorem 1, $J_{\lambda^2}(\mu, \nu)$ refers to the right-hand side of (4). More generally, the right derivative of $\sigma \mapsto J_\sigma$ exists for all $\sigma \geq 0$ and equals $\frac{1}{4}I_{\sqrt{\sigma}}(\mu, \nu)$.*

*Proof.* Since $\sigma \mapsto J_\sigma(\mu, \nu)$ is defined as an infimum of affine functions in $\sigma$, it is concave. Let $(\sigma_n)_{n \in \mathbb{N}}$ be a decreasing sequence of positive real numbers converging to 0 and let

$$\alpha_n = \frac{J_{\sigma_n} - J_0}{\sigma_n}.$$

By concavity, $\alpha_n$ is non-decreasing and admits a limit $J_0' = \frac{\mathrm{d}}{\mathrm{d}\sigma} J_\sigma|_{\sigma=0_+}$ that is the right derivative of $J$ at $0$. Our goal is to show that $J_0' = I_0(\mu,\nu)/4$. By the argument in the proof of Theorem (1), we have $\alpha_n \leq I_0(\mu,\nu)/4 \; \forall n$ thus $J_0' \leq I_0(\mu,\nu)/4$, so we just have to prove the other inequality.

Let $(\rho_n, v_n)_{n\geq 0}$ be a sequence of minimizers for the r.h.s. of Eq. (4) (which is in fact unique although we do not use that fact here [30]) with $\lambda^2 = \sigma_n$ and let $V_n = \int_0^1 \int_{\mathbb{R}^d} \|v_n\|_2^2 \mathrm{d}\rho_n$ and $I_n = \int_0^1 \int_{\mathbb{R}^d} \|\nabla \log(\rho_n)\|_2^2 \mathrm{d}\rho_n$. Since $V_n$ is uniformly bounded and converges to $V_0 = W_2^2(\mu,\nu)$, we have that $\rho_n$ converges weakly (in duality with continuous functions with compact support) to $\rho_0$, the unique constant speed Wasserstein geodesic between $\mu$ and $\nu$ (see, e.g. [18, Cor. 4.10] or by an application of [29, Proposition 2.2] as below). Moreover, since $V_n \geq V_0$, it holds

$$\alpha_n = \frac{V_n - V_0}{\sigma_n} + \frac{1}{4} I_n \geq \frac{1}{4} I_n$$

and in particular we have the uniform bound $I_n \leq I_0$. It follows by [29, Proposition 2.2] applied to the quantity $I_n = \int_0^1 \int_{\mathbb{R}^d} \|\frac{\mathrm{d}(\nabla \rho_n)}{\mathrm{d}\rho_n}\|_2^2 \mathrm{d}\rho_n(x,t)$ that $\nabla \rho_n$, seen as a vector valued measure on $[0,1] \times \mathbb{R}^d$, admits a weak limit denoted $\omega$ which is absolutely continuous with respect to $\rho_0$ and that $\liminf I_n \geq \int_0^1 \int_{\mathbb{R}^d} \|\frac{\mathrm{d}\omega}{\mathrm{d}\rho_0}\|_2^2 \mathrm{d}\rho_0(t,x)$. Since for any compactly supported function $\varphi \in \mathcal{C}^1([0,1] \times \mathbb{R}^d; \mathbb{R}^d)$ it holds $\int \mathrm{div}_x(\varphi) \mathrm{d}\rho_n \to \int \mathrm{div}_x(\varphi) \mathrm{d}\rho_0$ and $\int \varphi \cdot \mathrm{d}(\nabla \rho_n) \to \int \varphi \cdot \mathrm{d}\omega$, we have that $\omega = \nabla_x \rho_0$ and thus the previous integral is precisely the Fisher information of $\rho_0$ integrated in time. It follows that $\liminf I_n \geq I_0$ hence $J_0' \geq \frac{1}{4} I_0$ which concludes the proof. Inspecting the above argument, we see that in fact it applies directly to the case $\sigma > 0$ (except that of course the trajectory recovered as $n \to \infty$ is $\rho_{\sqrt{\sigma}}$), hence our second claim. $\qquad\square$

**Bounds on the Fisher information of the geodesic.** Let us now prove the bounds on the Fisher information of the Wasserstein geodesic that appear in Proposition 1. The main idea is to express $I_0(\mu,\nu)$ in terms of the initial and final densities and the Brenier potential.

*Proof of Proposition 1.* Let us express $I_0(\mu,\nu)$ in terms of the densities $\rho_0$ and $\rho_1$ (of $\mu$ and $\nu$ respectively) and the Brenier potential $\varphi$ which is the convex function such that $(\nabla\varphi)_{\#}\mu = \nu$, i.e. $\nu$ is the pushforward of $\mu$ by the map $\nabla\varphi$. Let $(\rho_t)_{t\in[0,1]}$ be the density of the $W_2$-geodesic between $\mu$ and $\nu$. We start with the conservation of mass formula which holds under our regularity assumptions:

$$\rho_0(x) = \det(\nabla^2 \varphi_t(x)) \rho_t(\nabla \varphi_t(x)))$$

where $\varphi_t(x) \overset{\text{def.}}{=} (1-t)\|x\|_2^2/2 + t\varphi(x)$ is such that $(\nabla\varphi_t)_{\#}\rho_0 = \rho_t$. By taking the logarithm we get

$$\log \rho_0(x) = \log \rho_t(\nabla \varphi_t(x)) + \log \det(\nabla^2 \varphi_t(x)).$$

Let us now take the gradient of this expression. We denote by $d^3\varphi(x) : \mathbb{R}^d \to \mathbb{R}^{d\times d}$ the weak differential of $x \mapsto \nabla^2\varphi(x)$ (which exists for almost every $x$ and is bounded since $\nabla^2\varphi$ is assumed Lipschitz) and by $[d^3\varphi(x)]^* : \mathbb{R}^{d\times d} \to \mathbb{R}^d$ its adjoint. Using the fact that the differential of $A \mapsto \log \det A$ at $A$ is the scalar product with $A^{-1}$ we get that for almost every $x \in \mathbb{R}^d$,

$$\nabla \log \rho_0(x) = \nabla^2 \varphi_t(x) \nabla \log \rho_t(\nabla \varphi_t(x)) + [d^3\varphi_t(x)]^* [\nabla^2 \varphi_t(x)]^{-1}. \tag{7}$$

It follows that

$$I_0(\mu,\nu) = \int_0^1 \int_{\mathbb{R}^d} \|\nabla \log \rho_t(x)\|_2^2 \rho_t(x) \mathrm{d}x \mathrm{d}t$$

$$= \int_0^1 \int_{\mathbb{R}^d} \|\nabla \log \rho_t(\nabla \varphi_t(x))\|_2^2 \rho_0(x) \mathrm{d}x \mathrm{d}t$$

$$= \int_0^1 \int_{\mathbb{R}^d} \|[\nabla^2 \varphi_t(x)]^{-1} \nabla \log \rho_0(x) - t[\nabla^2 \varphi_t(x)]^{-1}[d^3\varphi(x)]^*[\nabla^2 \varphi_t(x)]^{-1}\|_2^2 \rho_0(x) \mathrm{d}x \mathrm{d}t$$

where we have used the fact that $d^3\varphi_t(x) = t d^3\varphi(x)$.

**General case.** In the general case, we simply use the bounds $\nabla^2 \varphi_t(x) \succeq ((1-t) + t\kappa)\mathrm{Id}$ and $\|d^3\varphi(x)\| \leq L$ almost everywhere in operator norm and the identity $|a+b|^2 \leq 2|a|^2 + 2|b|^2$ valid for any $a, b \in \mathbb{R}$ to get

$$I_0(\mu, \nu) \leq 2\Big( \int_0^1 \frac{\mathrm{d}t}{(1 + (\kappa - 1))^2} \Big) I_0(\mu, \mu) + 2\Big( \int_0^1 \frac{t^2 \mathrm{d}t}{(1 + (\kappa - 1))^4} \Big) L^2$$
$$= 2\kappa^{-1} I_0(\mu, \mu) + (2/3)\kappa^{-3} L^2.$$

**One dimensional case.** When $d = 1$, from Eq. (7) at time $t = 1$, we get
$$\varphi'''(x) = \nabla \log \rho_0(x) \varphi''(x) - \nabla \rho_1(\nabla \varphi(x)) \varphi''(x)^2.$$

Plugging this expression in the previous integral leads to:

$$I_0(\mu, \nu) = \int_{\mathbb{R}} \int_0^1 \left( \frac{(1-t)\nabla \log \rho_0 + t \nabla \log \rho_1(\nabla \varphi(x)) \varphi''(x)^2}{\big((1-t) + t\varphi''(x)\big)^2} \right)^2 \mathrm{d}t \rho_0(x) \mathrm{d}x$$

With the valid change of variables $1 - s = \frac{t\varphi''(x)}{(1-t)+t\varphi''(x)}$ (and thus $s = \frac{1-t}{(1-t)+t\varphi''(x)}$), we obtain:

$$I_0(\mu, \nu) = \int_{\mathbb{R}} \int_0^1 \frac{1}{\varphi''(x)} \Big( (1-s)\nabla \log \rho_0(x) + s \nabla \log \rho_1(\varphi'(x)) \varphi''(x) \Big)^2 \mathrm{d}s \rho_0(x) \mathrm{d}x$$
$$= \int_{\mathbb{R}} \int_0^1 \frac{1}{\varphi''(x)} \Big( (1-s)\nabla \log \rho_0(x) + s \nabla \log(\rho_1 \circ \varphi')(x) \Big)^2 \mathrm{d}s \rho_0(x) \mathrm{d}x$$

This leads to the bound $I_0(\mu, \nu) \leq (2/3)\kappa^{-1} I_0(\mu, \mu) + (2/3)K I_0(\nu, \nu)$ since $(\varphi')_\# \mu = \nu$. $\qquad \square$

**Gaussian case.** Let us now give the explicit expression of the Fisher information of the Wasserstein geodesic between Gaussian distributions, which is mentioned in Section 4. Whenever we deal with a positive semidefinite matrix $A$, the matrix $A^{1/2}$ refers to its unique positive semidefinite square root.

**Proposition 9.** *If $\mu = \mathcal{N}(0, A)$, $\nu = \mathcal{N}(0, B)$ then $I_0(\mu, \nu) = \mathrm{tr}\, S^{-1}$ with $S = (A^{1/2} B A^{1/2})^{1/2}$.*

Remark in particular that the expansion in Theorem 1 then gives

$$S_\lambda(\mu, \nu) - W_2^2(\mu, \nu) = \frac{1}{8}(2I_0(\mu, \nu) - I_0(\mu, \mu) - I_0(\nu, \nu)) = \frac{1}{8}(2\,\mathrm{tr}\, S^{-1} - \mathrm{tr}\, A^{-1} - \mathrm{tr}\, B^{-1})$$

which is consistent, as it must, with the expansion in Section 4.

*Proof.* When the Brenier potential $\varphi = \frac{1}{2} x^\top H x$ is quadratic, we have by the proof of Proposition 1

$$I_0(\mu, \nu) = \int_{\mathbb{R}^d} \int_0^1 \|[\nabla^2 \varphi_t(x)]^{-1} \nabla \log \rho_0(x)\|_2^2 \rho_0(x) \mathrm{d}t \mathrm{d}x.$$

Putting ourselves in a basis diagonalizing $H$, the integration in time is explicit and we get

$$I_0(\mu, \nu) = \int_{\mathbb{R}^d} \|H^{-1/2} \nabla \log \rho_0(x)\|_2^2 \rho_0(x) \mathrm{d}x.$$

It turns out that if $\mu = \mathcal{N}(0, A)$, $\nu = \mathcal{N}(0, B)$, then $\varphi(x) = \frac{1}{2} x^T H x$ where [6]

$$H = A^{-1/2} \Big( A^{1/2} B A^{1/2} \Big)^{1/2} A^{-1/2}$$

and thus

$$I_0(\mu, \nu) = \int_{\mathbb{R}^d} \|H^{-1/2} \nabla \log \rho_0(x)\|_2^2 \rho_0(x) \mathrm{d}x.$$
$$= \int_{\mathbb{R}^d} \|H^{-1/2} A^{-1} x\|_2^2 \rho_0(x) \mathrm{d}x$$
$$= \mathbf{E}_{X \sim \mathcal{N}(0, A)} \Big[ X^T \big( A^{-1} H^{-1} A^{-1} \big) X \Big]$$
$$= \mathrm{tr}\, \big( A^{-1} H^{-1} A^{-1} A \big) = \mathrm{tr}\, \big( A^{-1} H^{-1} \big) = \mathrm{tr}\, \big( (A^{1/2} B A^{1/2})^{-1/2} \big)$$

where the last row is obtained using [45, Eq. (378)]. $\qquad \square$

# B  Computational complexity of Sinkhorn's algorithm

In this appendix, we recall the computational complexity analysis of Sinkhorn's algorithm from [20], in order to state Proposition 11 exactly as per our needs (while this result is implicit in [20]). There is nothing specific in this analysis about the squared-distance cost so we just assume that the cost $c : \mathbb{R}^d \times \mathbb{R}^d \to \mathbb{R}$ is continuous, keeping in mind that in our case, $c(x,y) = \frac{1}{2}\|y - x\|_2^2$. We also consider a compact set $\mathcal{X} \subset \mathbb{R}^d$ and measures $\mu, \nu \in \mathcal{P}(\mathcal{X})$ which are concentrated on this set. We consider the dual objective function of entropy regularized optimal transport [46]:

$$F_\lambda(u,v) = \int_{\mathbb{R}^d} u \, \mathrm{d}\mu + \int_{\mathbb{R}^d} v \, \mathrm{d}\nu + \lambda\Big(1 - \int_{(\mathbb{R}^d)^2} \exp((u(x) + v(y) - c(x,y))/\lambda)\mathrm{d}\mu(x)\mathrm{d}\nu(y)\Big). \quad (8)$$

By Fenchel duality, we have with $c(x,y) = \frac{1}{2}\|y - x\|_2^2$ that

$$\frac{1}{2}T_\lambda(\mu,\nu) = \max_{u,v} F_\lambda(u,v) \quad (9)$$

where the maximum is over pairs of continuous real-valued functions on $\mathbb{R}^d$, $(u,v) \in \mathcal{C}(\mathcal{X})^2$. Sinkhorn's algorithm is alternate maximization on $u$ and $v$: it starts with $u_0, v_0 = 0$ and defines,

$$u_{k+1} = u_k - \lambda \log \int_{\mathbb{R}^d} \exp((u_k(\cdot) + v_k(y) - c(\cdot,y))/\lambda)\mathrm{d}\nu(y), \quad v_{k+1} = v_k \quad \text{if } k \text{ is odd}$$

$$v_{k+1} = v_k - \lambda \log \int_{\mathbb{R}^d} \exp((u_k(x) + v_k(\cdot) - c(x,\cdot))/\lambda)\mathrm{d}\mu(x), \quad u_{k+1} = u_k \quad \text{if } k \text{ is even.}$$

This form of the iterations that distinguishes between even and odd updates is convenient for the analysis, but beware that the index $k$ here is twice the index appearing in Proposition 2, so the statements are adjusted consequently. We also introduce $\gamma_k = \exp((u_k(x) + v_k(y) - c(x,y))/\lambda)\mu \otimes \nu$, which is such that the update can be written: $u_{k+1} = u_k + \lambda \log(\mathrm{d}\mu/\mathrm{d}\pi_\#^1 \gamma_k)$ if $k$ odd and $v_{k+1} = v_k + \lambda \log(\mathrm{d}\nu/\mathrm{d}\pi_\#^2 \gamma_k)$ if $k$ even, where $\pi_\#^1 \gamma$ is the marginal of $\gamma$ on the first factor of $\mathbb{R}^d \times \mathbb{R}^d$ and $\pi_\#^2 \gamma$ its marginal on the second. The following is a rearrangement of some intermediate results in [20] in a simplified form which is sufficient to our purpose.

**Proposition 10.** *Assume $c \geq 0$ and let $\|c\|_\infty = \sup_{(x,y)\in\mathcal{X}^2} c(x,y)$. Sinkhorn's iterates satisfy, for $k \geq 1$,*

$$0 \leq \max_{u,v} F_\lambda(u,v) - F_\lambda(u_k, v_k) \leq \frac{2\|c\|_\infty^2}{\lambda k}$$

*Proof.* First, remark that the iterations are such that $\int \mathrm{d}\gamma_k = 1$ for $k \geq 1$, so it holds $F_\lambda(u,v) = \int u \, \mathrm{d}\mu + \int v \, \mathrm{d}\nu$ for $(u,v) = (u_k, v_k)$ and also for any maximizer $(u,v) = (u^*, v^*)$. The key of the proof is the following equality first noticed by [2]. If $k$ is odd, then

$$F_\lambda(u_{k+1}, v_{k+1}) - F_\lambda(u_k, v_k) = -\lambda \int \log\Big(\int \exp((u_k(x) + v_k(y) - c(x,y))/\lambda)\mathrm{d}\nu(y)\Big)\mathrm{d}\mu(x)$$

$$= \lambda \int \log(\mathrm{d}\mu/\mathrm{d}\pi_\#^1 \gamma_k)\mathrm{d}\mu = \lambda H(\mu, \pi_\#^1 \gamma_k).$$

Let us define $\Delta_k = F_\lambda(u^*, v^*) - F_\lambda(u_k, v_k) \geq 0$. Using Pinsker's inequality and Lemma 2 it follows

$$\Delta_k - \Delta_{k+1} \geq \frac{\lambda}{2}\|\mu - \pi_\#^1 \gamma_k\|_1^2 \geq \frac{\lambda}{2\|c\|_\infty^2}\Delta_k^2.$$

We can similarly prove the same inequality for $k$ even. We conclude as in the usual proof of gradient descent for smooth functions [43, Thm. 2.1.14]: by dividing by $\Delta_k \Delta_{k+1}$ we have

$$\frac{1}{\Delta_{k+1}} - \frac{1}{\Delta_k} \geq \frac{\lambda}{2\|c\|_\infty^2}\frac{\Delta_k}{\Delta_{k+1}} \geq \frac{\lambda}{2\|c\|_\infty^2}.$$

Summing these inequalities yields a telescopic sum and we get $1/\Delta_k \geq \lambda k/(2\|c\|_\infty^2)$ which allows to conclude. $\qquad\square$

From this analysis, we deduce the following complexity to approximate $T_\lambda$ and $T_0$ using Sinkhorn's iterations, adapted from [20].

**Proposition 11.** *Assume that $\mu_n = \sum_{i=1}^n p_i \delta_{x_i}$ and $\nu_n = \sum_{j=1}^n q_j \delta_{y_j}$ are discrete measures with $n$ atoms such that $p_i, q_j \geq \alpha/n$ for some $\alpha > 0$. Then Sinkhorn's algorithm returns an $\varepsilon$-accurate estimation of $T_\lambda(\mu, \nu)$ in time $O(n^2 \|c\|_\infty^2 / (\lambda \varepsilon))$. Moreover, fixing $\lambda = \varepsilon/4(\log(n) + \log(1/\alpha))$, it returns an $\varepsilon$-accurate estimation of $T_0(\mu, \nu)$ in $O(n^2 \log(n) \|c\|_\infty^2 / \varepsilon^2)$ operations.*

*Proof.* The first claim is a direct consequence of Proposition 10 since when $\mu$ and $\nu$ have a finite support of size $n$, an iteration of Sinkhorn can be performed with $O(n^2)$ operations. The second claim follows from the bound

$$0 \leq T_\lambda(\mu, \nu) - T_0(\mu, \nu) \leq 2\lambda H(\gamma^*, \mu \otimes \nu) \leq 4\lambda(\log n + \log(1/\alpha))$$

where $\gamma^*$ is the optimal transport plan for $T_0$. $\square$

**Lemma 2.** *Under the assumptions and notations of Proposition 10 it holds*

$$\Delta_k \leq \|c\|_\infty \Big( \|\mu - \pi^1_\# \gamma_k\|_1 + \|\nu - \pi^2_\# \gamma_k\|_1 \Big)$$

*where $\|\mu\|_1 \overset{\text{def.}}{=} \sup_{\|u\|_\infty \leq 1} \int u(x) \mathrm{d}\mu(x)$ denotes the total variation norm in the space of measures.*

*Proof.* Remark that $F_\lambda$ is differentiable in $(u, v)$ with gradient $(\mu - \pi^1_\# \gamma_k, \nu - \pi^2_\# \gamma_k)$ at $(u_k, v_k)$. The concavity inequality then gives

$$\Delta_k \leq \int (u^* - u_k) \mathrm{d}(\mu - \pi^1_\# \gamma_k) + \int (v^* - v_k) \mathrm{d}(\nu - \pi^2_\# \gamma_k).$$

Also, for any $u \in \mathcal{C}(\mathcal{X})$ and $\alpha = (\max u + \min u)/2$, using the fact that $\int \mu = \int \pi^1_\# \gamma_k$, we have

$$\int u \mathrm{d}(\mu - \pi^1_\# \gamma_k) = \int (u - \alpha) \mathrm{d}(\mu - \pi^1_\# \gamma_k) \leq \frac{1}{2} (\max u - \min u) \|\mu - \pi^1_\# \gamma_k\|_1.$$

Finally, for $u = u^*$ or $u = u_k$ for $k \geq 1$, we have, for some $v \in \mathcal{C}(\mathcal{X})$, and for all $x, x' \in \mathcal{X}$

$$u(x) = -\lambda \log \int \exp((v(y) - c(x, y))/\lambda) \mathrm{d}\nu(y) \leq \|c\|_\infty - u(x')$$

because $c(x, y) \leq c(x', y) + \|c\|_\infty$. Thus $(\max u - \min u)/2 \leq \|c\|_\infty/2$. The conclusion follows by bounding all terms this way. $\square$

## C   Properties of the plug-in estimator

In this section we prove Theorem 2 about the rate of convergence of $T_0(\hat\mu_n, \hat\nu_n)$ to $T_0(\mu, \nu)$ (we recall that, by definition $W_2^2(\mu, \nu) = T_0(\mu, \nu)$). We start with the following lemma which bounds the estimation error by simpler quantities. Note that we consider measures on the centered ball of radius $R$ in $\mathbb{R}^d$, for some $R > 0$, which is without loss of generality compared to other bounded sets since $T_\lambda(\mu, \nu)$ is invariant by translation of both measures. In the following lemma $\mu_n, \nu_n \in \mathcal{P}(\mathbb{R}^d)$ can be unrelated to $\mu, \nu$ but this lemma will later be applied to the case where $\mu_n, \nu_n$ are empirical distributions of $n$ samples, hence our choice of notation.

**Lemma 3.** *Let $\mu, \nu, \mu_n, \nu_n \in \mathcal{P}(\mathbb{R}^d)$ be concentrated on the centered ball of radius $R$. Then it holds*

$$\left| \frac{1}{2} T_0(\mu, \nu) - \frac{1}{2} T_0(\mu_n, \nu_n) \right| \leq \left| \frac{1}{2} \int \|x\|_2^2 \mathrm{d}(\mu - \mu_n)(x) \right| + \left| \frac{1}{2} \int \|x\|_2^2 \mathrm{d}(\nu - \nu_n)(x) \right|$$

$$+ \sup_{\varphi \in \mathcal{F}_R} \left| \int \varphi \mathrm{d}(\mu_n - \mu) \right| + \sup_{\varphi \in \mathcal{F}_R} \left| \int \varphi \mathrm{d}(\nu_n - \nu) \right|$$

*where $\mathcal{F}_R$ is the set of convex and $R$-Lipschitz functions on the ball of radius $R$.*

*Proof.* The first part of the proof is fairly classical. By Kantorovich duality, we have

$$\frac{1}{2}T_0(\mu,\nu) = \max_{u,v \in \mathcal{C}(\mathcal{X})} \int u \mathrm{d}\mu + \int v \mathrm{d}\nu$$

where $\mathcal{X}$ is the closed ball of radius $R$ and under the constraint that $u(x) + v(y) \leq \frac{1}{2}\|y-x\|_2^2$ for all $(x,y) \in \mathcal{X}^2$ and there exists a maximizer [52]. By expanding the square and changing the unknown $(\varphi,\psi) = (\frac{1}{2}\|\cdot\|_2^2 - u, \frac{1}{2}\|\cdot\|_2^2 - v)$, we can equivalently write

$$\frac{1}{2}T_0(\mu,\nu) = \frac{1}{2}\int \|x\|_2^2 \mathrm{d}\mu(x) + \frac{1}{2}\int \|x\|_2^2 \mathrm{d}\nu(x) - \min_{\varphi,\psi \in \mathcal{C}(\mathcal{X})} \left(\int \varphi \mathrm{d}\mu + \int \psi \mathrm{d}\nu\right)$$

under the constraint that $\varphi(x) + \psi(y) \geq \langle x,y \rangle$ for all $(x,y) \in \mathcal{X}^2$. In the minimization problem, fix an arbitrary $\psi \in \mathcal{C}(\mathcal{X})$ and notice that the value of the objective cannot increase if we replace $\varphi$ by $\psi^*$ defined by $\psi^*(x) = \max_{y \in \mathcal{X}} \langle x,y \rangle - \psi(y)$ and the couple $(\psi^*, \psi)$ still satisfies the constraint. Repeating this process by now fixing $\psi^*$, we find that the couple $(\psi^*, \psi^{**})$ satisfies the constraint and has a smaller objective value. Now, as a supremum of affine functions, $\psi^*$ is convex. For any $y_0 \in \mathcal{X}$, let $x_0$ be such that $\psi^*(y_0) = \langle x_0, y_0 \rangle - \psi(x_0)$, and observe that for all $y \in \mathcal{X}$

$$\begin{cases} \psi^*(y_0) = \langle x_0, y_0 \rangle - \psi(y_0) \\ \psi^*(y) \geq \langle x_0, y \rangle - \psi(y) \end{cases} \quad \Rightarrow \quad \psi^*(y_0) - \psi^*(y) \leq \langle x_0, y_0 - y \rangle \leq R\|y_0 - y\|_2.$$

Since $y_0$ and $y$ are arbitrary, this shows that $\psi^*$ is $R$-Lipschitz, i.e., $|\psi^*(y) - \psi^*(y')| \leq R\|y - y'\|_2$ for all $(y,y') \in \mathcal{X}^2$. We thus have

$$\frac{1}{2}T_0(\mu,\nu) = \frac{1}{2}\int \|x\|_2^2 \mathrm{d}\mu(x) + \frac{1}{2}\int \|x\|_2^2 \mathrm{d}\nu(x) - \min_{\varphi \in \mathcal{F}_{\mathcal{R}}} \left(\int \varphi \mathrm{d}\mu + \int \varphi^* \mathrm{d}\nu\right).$$

The rest of the proof is inspired by [40, Prop. 2] (which analyzes the sample complexity of $T_\lambda$ for $\lambda > 0$). Let us denote $\mathcal{S}_{\mu,\nu}(\varphi) \overset{\text{def}}{=} \int \varphi \mathrm{d}\mu + \int \varphi^* \mathrm{d}\nu$ and $\varphi_{\mu,\nu}$ the minimizer of $\mathcal{S}_{\mu,\nu}$ over $\mathcal{F}_R$. By optimality, we have

$$\mathcal{S}_{\mu,\nu}(\varphi_{\mu_n,\nu}) - \mathcal{S}_{\mu_n,\nu}(\varphi_{\mu_n,\nu}) \leq \mathcal{S}_{\mu,\nu}(\varphi_{\mu,\nu}) - \mathcal{S}_{\mu_n,\nu}(\varphi_{\mu_n,\nu}) \leq \mathcal{S}_{\mu,\nu}(\varphi_{\mu,\nu}) - \mathcal{S}_{\mu_n,\nu}(\varphi_{\mu,\nu}).$$

It follows that

$$|\mathcal{S}_{\mu,\nu}(\varphi_{\mu,\nu}) - \mathcal{S}_{\mu_n,\nu}(\varphi_{\mu_n,\nu})| \leq \max\left\{|\mathcal{S}_{\mu,\nu}(\varphi_{\mu_n,\nu}) - \mathcal{S}_{\mu_n,\nu}(\varphi_{\mu_n,\nu})|, |\mathcal{S}_{\mu,\nu}(\varphi_{\mu,\nu}) - \mathcal{S}_{\mu_n,\nu}(\varphi_{\mu,\nu})|\right\}$$

$$\leq \sup_{\varphi \in \mathcal{F}_R} |\mathcal{S}_{\mu,\nu}(\varphi) - \mathcal{S}_{\mu_n,\nu}(\varphi)| = \sup_{\varphi \in \mathcal{F}_R} \left|\int \varphi \mathrm{d}(\mu_n - \mu)\right|.$$

As a consequence, we have

$$\left|\frac{1}{2}T_0(\mu,\nu) - \frac{1}{2}T_0(\mu_n,\nu)\right| \leq \left|\frac{1}{2}\int \|x\|_2^2 \mathrm{d}(\mu - \mu_n)(x)\right| + \sup_{\varphi \in \mathcal{F}_R} \left|\int \varphi \mathrm{d}(\mu_n - \mu)\right|.$$

We finally conclude with the triangle inequality

$$|T_0(\mu,\nu) - T_0(\mu_n,\nu_n)| \leq |T_0(\mu,\nu) - T_0(\mu_n,\nu)| + |T_0(\mu_n,\nu) - T_0(\mu_n,\nu_n)|$$

and by bounding the second term in the same fashion. $\qquad\square$

The next technical step is to bound the supremum of an empirical process that appears in the bound of Lemma 3.

**Lemma 4.** *Let $\mu \in \mathcal{P}(\mathbb{R}^d)$ be concentrated on the ball of radius $R$ and $\hat{\mu}_n$ an empirical distribution of $n$ independent samples. Then it holds*

$$\mathbf{E}\left[\sup_{\varphi \in \mathcal{F}_R} \left|\int \varphi \mathrm{d}(\hat{\mu}_n - \mu)\right|\right] \lesssim \begin{cases} R^2 n^{-1/2} & \text{if } d < 4, \\ R^2 n^{-1/2} \log(n) & \text{if } d = 4, \\ R^2 n^{-2/d} & \text{if } d > 4 \end{cases}$$

*where the notation $\lesssim$ hides a constant depending only on $d$ and $\mathcal{F}_R$ is defined in Lemma 3.*

*Proof.* First notice that we can include in the definition of $\mathcal{F}_R$ the property that $\varphi(0) = 0$ without changing the supremum. With this additional property, we in particular have that $\|\varphi\|_\infty \leq R^2$ for all $\varphi \in \mathcal{F}_R$. By a classical symmetrization argument [58, Thm. 4.10], we have

$$\mathbf{E}\Big[\sup_{\varphi \in \mathcal{F}_R}\Big|\int \varphi\mathrm{d}(\hat{\mu}_n - \mu)\Big|\Big] \leq 2\underbrace{\mathbf{E}_{\sigma,X}\Big[\sup_{\varphi \in \mathcal{F}_R}\Big|\frac{1}{n}\sum_{i=1}^n \sigma_i\varphi(X_i)\Big|\Big]}_{\mathcal{R}_n(\mathcal{F}_R,\mu)}$$

where $\sigma_1, \ldots, \sigma_n$ are independent Rademacher random variables taking the values $\{-1, +1\}$ with equal probability and $X_1, \ldots, X_n$ are independent random variables with law $\mu$. This quantity $\mathcal{R}_n(\mathcal{F}_R, \mu)$ is the Rademacher complexity of $\mathcal{F}_R$ under the distribution $\mu$. It can be bounded by Dudley's chaining technique (see [58, Thm. 5.22] and the associated discussion): it holds, for some universal constant $C > 0$,

$$\mathcal{R}_n(\mathcal{F}_R,\mu) \leq C\inf_{\delta>0}\Big(\delta + n^{-1/2}\int_\delta^{R^2}\sqrt{\log N_\infty(\mathcal{F}_R,u)}\mathrm{d}u\Big)$$

where $N_\infty(\mathcal{F}_R, u)$ is the covering number of the set $\mathcal{F}_R$ for the metric $\|\cdot\|_\infty$ at scale $u$. Then we use the covering number bound of Bronshtein [10], as reported in [32, Thm. 1] which states that there exists constants $C_1, C_2 > 0$ depending only on $d$ such that if $u/R^2 \leq C_1$ then

$$\log N_\infty(\mathcal{F}_R,u) \leq C_2(u/R^2)^{-d/2}.$$

After a change of variable we thus have that

$$\mathcal{R}_n(\mathcal{F}_R,\mu) \lesssim R^2\Big(\inf_{\delta>0}\delta + n^{-1/2}\int_\delta^1 u^{-d/4}\mathrm{d}u\Big).$$

The claim follows by optimizing over $\delta$ which gives $\delta = 0$ for $d < 4$, $\delta = n^{-1/2}$ for $d = 4$ and $\delta = n^{-2/d}$ for $d > 4$. $\qquad\square$

We are now in position to conclude the proof of Theorem 2.

*Proof of Theorem 2.* Let us assume without loss of generality that $\mu, \nu$ are concentrated on the centered closed ball of radius $R$ in $\mathbb{R}^d$ (which can be taken as $R = 1/2$ under our assumptions, but let us continue with an arbitrary $R$ for explicitness of the proof). Given Lemma 3 and Lemma 4, it only remains to bound the quantity

$$A \stackrel{\text{def.}}{=} \mathbf{E}\Big|\frac{1}{2}\int \|x\|_2^2\mathrm{d}(\mu - \hat{\mu}_n)(x)\Big|$$

and the corresponding quantity for $\nu$. Considering independent samples of the random variable $Y = \frac{1}{2}\|X\|_2^2$ where the law of $X$ is $\mu$, our goal is to bound $A = \mathbf{E}|\frac{1}{n}\sum_{i=1}^n Y_i - \mathbf{E}Y|$. By Chebyshev's inequality and the fact that the variance of $Y$ is bounded by $R^4$, we have for all $t \geq 0$,

$$\mathbf{P}\Big[\big|\frac{1}{n}\sum_{i=1}^n Y_i - \mathbf{E}Y\big| \geq t\Big] \leq \min\{1, R^4/(nt^2)\}.$$

Finally, by the integral representation of the expectation of a nonnegative random variable we have

$$A = \int_0^\infty \mathbf{P}\Big[\big|\frac{1}{n}\sum_{i=1}^n Y_i - \mathbf{E}Y\big| \geq t\Big]\mathrm{d}t \leq \frac{R^2}{\sqrt{n}} + \int_{R^2 n^{-1/2}}^\infty \frac{R^4}{nt^2}\mathrm{d}t = 2R^2 n^{-1/2}$$

which is sufficient to conclude. The concentration bound is proved separately in Proposition 12. $\quad\square$

Let us now prove the concentration bound, which is a consequence of the bounded difference inequality. We give a unified proof for $T_\lambda$ and $T_0$ since the argument is similar. The result for $T_0$ was known [59] but we are not aware of a similar result for $\lambda > 0$ (note that the concentration bound in [26] has an undesirable exponential dependency in $\lambda$ and the central limit theorem in [40] does not a priori gives the dependency in $\lambda$).

**Proposition 12.** *Assume that $\mu, \nu \in \mathcal{P}(\mathbb{R}^d)$ are concentrated on a set of diameter $D$. It holds for all $t \geq 0$, $\lambda \geq 0$ and $n \geq 1$,*

$$\mathbf{P}\Big[\big|T_\lambda(\mu_n, \nu_n) - \mathbf{E}[T_\lambda(\mu_n, \nu_n)]\big| \geq t\Big] \leq 2\exp(-nt^2/D^4).$$

*Proof.* As in [59], we want to apply the bounded difference inequality but we study the stability of the primal problem (instead of the dual) in order to cover the regularized case painlessly. The empirical measures are of the form $\mu_n = \frac{1}{n}\sum_{i=1}^n \delta_{x_i}$ and $\nu_n = \frac{1}{n}\sum_{j=1}^n \delta_{y_j}$. Let $c \in \mathbb{R}^{n \times n}$ be the cost matrix with entries $c_{i,j} = \frac{1}{2}\|x_i - y_j\|_2^2$. With those notations, it holds

$$\frac{1}{2}T_\lambda(\mu_n, \nu_n) = \min \sum_{i,j} c_{i,j}P_{i,j} + \lambda \sum_{i,j} P_{i,j}\log(n^2 P_{i,j})$$

where the minimum is over matrices $P \in \mathbb{R}_+^{n \times n}$ such that $P\mathbf{1} = \mathbf{1}/n$ and $P^\top\mathbf{1} = \mathbf{1}/n$ (i.e. $nP$ is bistochastic). Let $P^*$ be a minimizer. Now let $\tilde{\mu}_n = \frac{1}{n}(\sum_{i=1}^{n-1}\delta_{x_i} + \delta_{\tilde{x}_n})$ for some $\tilde{x}_i$ in the same set of diameter $D$. This changes one row in the cost matrix, each entry in this row being changed by at most $D^2/2$. Thus using $P^*$ as a candidate in the minimization problem defining $T_\lambda(\tilde{\mu}_n, \nu_n)$ we get $T_\lambda(\tilde{\mu}_n, \nu_n) \leq T_\lambda(\mu_n, \nu_n) + \frac{D^2}{n}$. Interchanging the role of $\mu_n$ and $\tilde{\mu}_n$, we get the reverse inequality and thus

$$|T_\lambda(\tilde{\mu}_n, \nu_n) - T_\lambda(\mu_n, \nu_n)| \leq \frac{D^2}{n}.$$

The same stability can be shown about perturbing $\nu_n$ by one sample. The proposition follows by applying the bounded difference inequality [58, Cor. 2.21], paying attention to the fact that the total number of samples is $2n$. $\qquad\square$

Finally, let us give the details of the proof of Proposition 3.

*Proof of Proposition 3.* By the concentration result we have that with probability $1 - \delta$, $|W_2^2(\mu_n, \nu_n) - \mathbf{E}W_2^2(\mu_n, \nu_n)| \lesssim n^{-1/2}\sqrt{\log(2/\delta)}$. Let us break down the proof into three cases depending on the dimension $d$.

If $d < 4$, then by choosing $n \gtrsim \log(2/\delta)\varepsilon^{-2}$, the quantity $W_2^2(\mu_n, \nu_n)$ has the desired accuracy with probability $1 - \delta$. Also choosing $\lambda \lesssim \varepsilon/(2\log n)$ guarantees that $|\hat{T}_{\lambda,n} - \hat{W}_2^2| \lesssim \varepsilon$. Thus, the computational complexity is $O(n^2/(\lambda\varepsilon)) = \tilde{O}(\varepsilon^{-6})$.

If $d > 4$, we can choose $n \gtrsim \log(2/\delta)^{d/4}\varepsilon^{-d/2}$ to reach the desired accuracy, which leads to a computational complexity in $\tilde{O}(\varepsilon^{-d-2})$.

Finally if $d = 4$, we can choose $n$ such that $\varepsilon \asymp n^{-1/2}(\log(n) + \sqrt{\log(2/\delta)})$ which leads to a computational complexity in $O(n^2\log(n)\varepsilon^{-2}) = O(\varepsilon^{-6}(\log n + \sqrt{\log(2/\delta)})^4) = \tilde{O}(\varepsilon^{-6})$. $\qquad\square$

## D   Analysis of the Sinkhorn divergence estimator given samples

Let us first state a result on the sample complexity to estimate $S_\lambda$ with $\hat{S}_{\lambda,n}$ which is defined, given $x_1, \ldots, x_n$ i.i.d. samples from $\mu$ and $y_1, \ldots, y_n$ i.i.d. samples from $\nu$, as $\hat{S}_{\lambda,n} = S_\lambda(\hat{\mu}_n, \hat{\nu}_n)$ as in Eq. (2) where $\hat{\mu}_n = \frac{1}{n}\sum_{i=1}^n \delta_{x_i}$ and $\hat{\nu}_n = \frac{1}{n}\sum_{i=1}^n \delta_{y_i}$. Since the following result has not yet been stated in the precise form that we use, we give a short proof below. It essentially just requires to combine the results from [40] and [26].

**Lemma 5.** *Let $\mu, \nu \in \mathcal{P}(\mathbb{R}^d)$ be concentrated on a set of diameter 1, let $\hat{\mu}_n, \hat{\nu}_n$ be empirical distributions with $n$ independent samples and let $d' = 2\lfloor d/2 \rfloor$. Then*

$$\mathbf{E}\Big[|\hat{S}_{\lambda,n} - S_\lambda(\mu, \nu)|\Big] \lesssim (1 + \lambda^{-d'/2})n^{-1/2}$$

*where $\lesssim$ hides a constant that only depends on $d$.*

*Proof.* It has been shown in [40, Cor. 2], with a strategy similar to that employed in the end of the proof of Lemma 3, that

$$\left| \frac{1}{2} T_\lambda(\hat\mu_n, \hat\nu_n) - \frac{1}{2} T_\lambda(\mu, \nu) \right| \leq \sup_{f \in \mathcal{F}} \left| \int f \mathrm{d}(\hat\mu_n - \mu) \right| + \sup_{f \in \mathcal{F}} \left| \int f \mathrm{d}(\hat\nu_n - \nu) \right|$$

where $\mathcal{F}$ is any class of functions that is large enough to contain all the solutions to Eq. (9) for all pairs of measures $\mu, \nu \in \mathcal{P}(\mathbb{R}^d)$ concentrated on a set of diameter 1. It was shown in [26, Thm. 2] that $\mathcal{F}$ can be chosen as a ball in the Sobolev space $H^s$, $s \geq 1$ with diameter $C(1 + \lambda^{1-s})$ for some $C > 0$ that only depends on $d$ and $s$. In particular, for $s = d'/2 + 1$, $H^s$ is a reproducible kernel Hilbert space. Thus, using the notion of Rademacher complexity introduced in the proof of Lemma 4 and its bound for balls in reproducible kernel Hilbert spaces (as in [26, Prop. 2]), it follows

$$\mathbf{E}\left[ \sup_{f \in \mathcal{F}} \left| \int f \mathrm{d}(\hat\mu_n - \mu) \right| \right] \leq 2\mathcal{R}_n(\mathcal{F}, \mu) \lesssim (1 + \lambda^{-d'/2}) n^{-1/2}.$$

This is sufficient to bound the expected estimation error of $T_\lambda$. Let us now turn our attention to $\hat{S}_{\lambda,n}$. It holds

$$|\hat{S}_{\lambda,n} - S_\lambda(\mu, \nu)| \leq |T_\lambda(\hat\mu_n, \hat\nu_n) - T_\lambda(\mu, \nu)|$$
$$+ \frac{1}{2}|T_\lambda(\hat\mu_n, \hat\mu_n) - T_\lambda(\mu, \mu)| + \frac{1}{2}|T_\lambda(\hat\nu_n, \hat\nu_n) - T_\lambda(\nu, \nu)|.$$

The argument in [40] goes through for each term and it follows that $\hat{S}_{\lambda,n}$ admits the same statistical bound (up to a constant) than $\hat{T}_{\lambda,n}$. □

*Proof of Proposition 4.* Let $W_2^2 = W_2^2(\mu, \nu)$ and $S_\lambda = S_\lambda(\mu, \nu)$. We consider the following error decomposition:

$$\mathbf{E}\big[|\hat{S}_{\lambda,n} - W_2^2|\big] \leq \mathbf{E}\big[|\hat{S}_{\lambda,n} - S_\lambda|\big] + |S_\lambda - W_2^2| \lesssim (1 + \lambda^{-d'/2}) n^{-1/2} + \lambda^2$$

where the first bound is from Lemma 5 and the second bound is an assumption. We then optimize the bound in $\lambda$ which gives $\lambda \asymp n^{-1/(d'+4)}$ and an error in $n^{-2/(d'+4)}$. For the concentration bound, we use the argument in the proof of Proposition 12 in Appendix C. Observe that if only one of the samples drawn from $\mu$ is changed, the resulting change in $\hat{S}_{\lambda,n}$ is at most $2/n$ which leads to, by the bounded difference inequality,

$$\mathbf{P}\Big[\big|S_\lambda(\mu_n, \nu_n) - \mathbf{E}[S_\lambda(\mu_n, \nu_n)]\big| \geq t\Big] \leq 2\exp(-nt^2/4).$$

□

*Proof of Proposition 5.* For $\hat{S}_{\lambda,n}^{(k)}$ we consider the error decomposition

$$|\hat{S}_{\lambda,n}^{(k)} - W_2^2| \leq |\hat{S}_{\lambda,n}^{(k)} - \hat{S}_{\lambda,n}| + |\hat{S}_{\lambda,n} - \mathbf{E}[\hat{S}_{\lambda,n}]| + \mathbf{E}|\hat{S}_{\lambda,n} - S_\lambda| + |S_\lambda - W_2^2|.$$

Let us choose $\lambda \asymp n^{-1/(d'+4)}$ as in Proposition 4. By the concentration result of Proposition 4, we have that with probability $1 - \delta$, $|\hat{S}_{\lambda,n} - \mathbf{E}\hat{S}_{\lambda,n}| \lesssim n^{-1/2}\sqrt{\log(2/\delta)}$ and thus $|\hat{S}_{\lambda,n} - W_2^2| \lesssim n^{-2/(d'+4)} + n^{-1/2}\sqrt{\log(2/\delta)}$. Thus by choosing $n \gtrsim \log(2/\delta)\varepsilon^{-(d'+4)/2}$ the quantity $\hat{S}_{\lambda,n}$ has the desired accuracy with probability $1 - \delta$. It follows that the computational complexity is $O(n^2/(\lambda\varepsilon)) = \tilde{O}(\varepsilon^{-d'-5.5})$.

For the second claim, we just remark that $n^{-2/(d'+4)}$ dominates the rate of the plug-in estimator given in Theorem 2 for all $d$, so both estimators can achieve an error of this order. However the difference is that with $\hat{S}_{\lambda,n}$ a regularization level $\lambda \asymp \varepsilon^{-1/2}$ is sufficient while $\lambda \lesssim \varepsilon/\log(n)$ is required for $\hat{T}_{\lambda,n}$ to achieve this error $\varepsilon$. The time complexity bounds then follows by Proposition 2. □

# E    Analysis of deterministic discretization

In this section, we consider probability distributions on the torus $\mu, \nu \in \mathcal{P}(\mathbb{T}^d)$ with densities with respect to the Lebesgue measure (also denoted $\mu, \nu$) and $c(x, y) = \frac{1}{2}\|[y - x]\|_2^2$ which is half the squared distance on the torus. We denote $[x] = x + k_0$ where $k_0 \in \mathbb{Z}^d$ is such that $\|x + k\|_2$ is minimal ($k_0$ is unique Lebesgue almost everywhere). We denote by $(u_\lambda, v_\lambda)$ the couple of minimizers of Eq. (8) that are fixed points of Sinkhorn's iterations

$$u_\lambda(x) = -\lambda \log \int e^{(v_\lambda(y) - c(x,y))/\lambda} \mathrm{d}\nu(y), \quad v_\lambda(y) = -\lambda \log \int e^{(u_\lambda(x) - c(x,y))/\lambda} \mathrm{d}\mu(x) \quad (10)$$

and such that $u_\lambda(0) = 0$. These properties uniquely define $(u_\lambda, v_\lambda)$ and we consider $p_\lambda(x, y) = \exp\big((u_\lambda(x) + v_\lambda(y) - c(x,y))/\lambda\big)\mu(x)\nu(y)$ which is the unique solution to (1). The following lemma gives some regularity estimates on $p_\lambda$. What is required in its proof is regularity of the marginals and of the cost function (which we fix to be the half squared-norm cost for consistency).

**Lemma 6** (Regularity of $p_\lambda$). *Assume that $\mu, \nu \in \mathcal{P}(\mathbb{T}^d)$ admit $M$-Lipschitz continuous log-densities. Then for almost every $z \in (\mathbb{T}^d)^2$ it holds*

$$\|\nabla \log p_\lambda(z)\|_2 \leq 4\sqrt{d}\lambda^{-1} + 2M.$$

*Moreover, it holds for all $z, z' \in (\mathbb{T}^d)^2$*

$$|p_\lambda(z) - p_\lambda(z')| \leq (e^{(4\sqrt{d}\lambda^{-1} + M)\|[z - z']\|_2} - 1)p_\lambda(z).$$

*Proof.* By differentiating the definition of $p_\lambda$, we have for almost every $(x, y) \in (\mathbb{T}^d)^2$

$$\nabla_x \log p_\lambda(x, y) = \frac{1}{\lambda}(\nabla u_\lambda(x) - [x - y]) + \nabla m(x).$$

where $m$ is the log-density of $\mu$. By differentiating Eq. (10), we also have

$$\nabla u_\lambda(x) = \int [x - y]e^{(u_\lambda(x) + v_\lambda(y) - c(x,y))/\lambda} \mathrm{d}\nu(y)$$

and thus $\|\nabla u_\lambda(x)\|_2 \leq \sup_{y \in \mathbb{T}^d} \|[y - x]\|_2 = \sqrt{d}$. It follows that $\sup_{x,y \in \mathbb{T}^d} \|\nabla_x \log p_\lambda(x,y)\|_2 \leq \frac{2\sqrt{d}}{\lambda} + M$, from which we deduce the first bound by also taking into account the $\nabla_y$ component. Now let $\alpha = 4\sqrt{d}\lambda^{-1} + 2M$. By Grönwall's inequality, we have $e^{-\alpha\|[z - z']\|_2}p_\lambda(z) \leq p_\lambda(z') \leq e^{\alpha\|[z - z']\|_2}p_\lambda(z)$ for all $z, z' \in (\mathbb{T}^d)^2$. It follows that $|p_\lambda(z') - p_\lambda(z)| \leq \max\{e^{\alpha\|[z' - z]\|_2} - 1, 1 - e^{-\alpha\|[z' - z]\|_2}\}p_\lambda(z)$ which implies our claim. $\qquad\square$

For a measure $\mu \in \mathcal{P}(\mathbb{T}^d)$ we call $\mu_h$ its finite volume discretization at resolution $h = 1/m$ for $m \in \mathcal{N}$ on the grid $(\mathbb{Z}/m\mathbb{Z})^d$. It is built via the following process: let $q_h : \mathbb{T}^d \to \mathbb{T}^d$ be defined by $q_h(x_1, \ldots, x_d) = (\frac{1}{m}\lfloor mx_1 + 1/2\rfloor, \ldots, \frac{1}{m}\lfloor mx_d + 1/2\rfloor)$. It maps each point $x \in \mathbb{T}^d$ to its closest point on the grid $(\mathbb{Z}/m\mathbb{Z})^d$ (with some arbitrary rule for ties). Then let $\mu_h \overset{\text{def.}}{=} (q_h)_{\#}\mu$ which gives to each point in the grid the mass that $\mu$ gives to its surrounding cell. Also, let us label the points in $(\mathbb{Z}/m\mathbb{Z})^d$ from 1 to $n = m^d$ as $(x_i)_{i=1}^n$ (we also use the notation $y_i = x_i$) and let us call $Q_j \subset \mathbb{T}^d$ the set of points which are mapped to the point labeled by $j \in \{1, \ldots, n\}$. We also call $Q_{i,j} = Q_i \times Q_j \subset (\mathbb{T}^d)^2$. We now state and prove a result that is slightly more precise than Proposition 6 were we control the error made by replacing measures by their discretization in the estimation of $T_\lambda$.

**Proposition 13** (Stability under discretization). *Assume that $\mu, \nu \in \mathcal{P}(\mathbb{T}^d)$ admit $M$-Lipschitz continuous log-densities and let $C > 0$ be any constant. If $h(M + \lambda^{-1}) \leq C$ then*

$$-h^2(1 + M) \lesssim T_\lambda(\mu_h, \nu_h) - T_\lambda(\mu, \nu) \lesssim \min\{h, h^2(\lambda^{-1} + M + 1)\}$$

*where $\lesssim$ hides constants that only depend on $d$ and $C$.*

*Proof.* The principle of the proof is to build admissible transport plans for the continuous (resp. discrete) problem from an admissible transport plan for the discrete (resp. continuous) problem and to bound the associated primal objectives functions.

**From discrete to continuous plans.** Consider any $\gamma_h \in \Pi(\mu_h, \nu_h)$ and consider $\gamma \in \Pi(\mu, \nu)$ the (unique) measure with a constant density with respect to $\mu \otimes \nu$ on each cell $Q_{i,j}$ and such that $(q_h \otimes q_h)_{\#}\gamma = \gamma_h$ (see [26, Def. 1] for a detailed construction in $\mathbb{R}^d$). By construction, it holds $H(\gamma, \mu \otimes \nu) = H(\gamma_h, \mu_h \otimes \nu_h)$. Let us bound the difference $\Delta_{i,j} = \int_{Q_{i,j}} (\frac{1}{2}\|[y - x]\|_2^2 - \frac{1}{2}\|[x_i - y_j]\|_2^2) \mathrm{d}\gamma(x, y)$. For clarity, let us assume that $[x - y] = x - y$ for all $(x, y) \in Q_{i,j}$, the argument being the same in each cell. We start with a second order Taylor expansion of the cost (which is exact with our quadratic cost):

$$\frac{1}{2}\|y - x\|_2^2 - \frac{1}{2}\|x_i - y_j\|_2^2 = (x_i - y_j)^\top(x - x_i) + (y_j - x_i)^\top(y - y_i)$$
$$+ \frac{1}{2}\|x - x_i\|_2^2 + \frac{1}{2}\|y - y_i\|_2^2 - (x - x_i)^\top(y - y_i).$$

Integrating the terms in the second row over $Q_{i,j}$, we get a quantity bounded by $dh^2/2$. For the terms in the first row, we see that we have to bound integrals of the form $|\sum_j \int_{Q_{i,j}} (x_i - y_j)^\top(x - x_i)\mathrm{d}\gamma(x, y)| \leq \sqrt{d}|\int_{Q_i} (x - x_i)\mu(x)\mathrm{d}x|$. So let us consider specifically the following integral:

$$\Delta_i = \Big| \int_{Q_i} (x - x_i)\mu(x)\mathrm{d}x \Big|$$
$$= \Big| \int_{Q_i} (x - x_i)(\mu(x) - |Q_i|^{-1}\int_{Q_i} \mu(x')\mathrm{d}x')\mathrm{d}x \Big|$$
$$\leq \sqrt{d}h|Q_i|^{-1} \int_{Q_i^2} |\mu(x) - \mu(x')|\mathrm{d}x\mathrm{d}x'$$

where we used the fact that $x_i$ is the center of mass of $Q_i$ for the Lebesgue measure and we denoted $|Q_i|$ the Lebesgue measure of $Q_i$. Now, since $\log \mu$ is $M$-Lipschitz an application of Grönwall's inequality as in the proof of Lemma 6 shows that $|\mu(x) - \mu(x')| \leq (e^{M\|x - x'\|_2} - 1)\mu(x)$. It thus follows that

$$\Delta_i \leq \sqrt{d}h(e^{Mh\sqrt{d}} - 1)\mu(Q_i) \lesssim Mh^2\mu(Q_i).$$

Putting all the bounds together and summing over all cells $Q_{i,j}$ we get

$$\int_{(\mathbb{T}^d)^2} \Big(\frac{1}{2}\|[y - x]\|_2^2 - \frac{1}{2}\|[x_i - y_j]\|_2^2\Big)\mathrm{d}\gamma(x, y) \lesssim h^2(1 + M).$$

From this it follows that for $\lambda \geq 0$, we have $T_\lambda(\mu, \nu) - T_\lambda(\mu_h, \nu_h) \lesssim h^2(1 + M)$.

**From continuous to discrete plans.** Consider any $\gamma \in \Pi(\mu, \nu)$ and consider its discretization $\gamma_h = (q_h \otimes q_h)_{\#}\gamma$. By the "information processing inequality", it holds $H(\gamma_h, \mu_h \otimes \nu_h) \leq H(\gamma, \mu \otimes \nu)$. Also, since the cost function is $\sqrt{d}$-Lipschitz on $\mathbb{T}^d$, we have the naive discretization bound

$$\Big| \int_{(\mathbb{T}^d)^2} \frac{1}{2}\|x - y\|_2^2 \mathrm{d}(\gamma - \gamma_h)(x, y) \Big| \lesssim h.$$

This is sufficient to deduce that $T_\lambda(\mu_h, \nu_h) - T_\lambda(\mu, \nu) \lesssim h$ for all $\lambda \geq 0$. Let us see however that a finer discretization bound can be given when $\gamma$ is the optimal solution of the entropy regularized problem using the regularity shown in Lemma 6. We denote $z = (x, y) \in (\mathbb{T}^d)^2$ and $z_{i,j} = (x_i, y_i)$ and we have, by decomposing the error into a first and second order term as in the first part of the proof,

$$\Big| \int_{(\mathbb{T}^d)^2} \frac{1}{2}\|y - x\|_2^2 \mathrm{d}(\gamma(x, y) - \gamma_h(x, y)) \Big| = \Big| \sum_{i,j} \int_{Q_{i,j}} (\frac{1}{2}\|y - x\|_2^2 - \frac{1}{2}\|x_i - y_j\|_2^2)\mathrm{d}\gamma(x, y) \Big|$$
$$\lesssim \sum_{i,j} \Big| \int_{Q_{i,j}} (z - z_{i,j})p_\lambda(z)\mathrm{d}z \Big| + h^2.$$

It remains to estimate the integral terms as can be done as in the first part of the proof by using the regularity of $\log p_\lambda$ given by Lemma 6

$$\Big| \int_{Q_{i,j}} (z - z_{i,j})p_\lambda(z)\mathrm{d}z \Big| \lesssim h|Q_{i,j}|^{-1} \int_{Q_{i,j}} \int_{Q_{i,j}} |p_\lambda(z) - p_\lambda(z')|\mathrm{d}z\mathrm{d}z'$$
$$\leq h(e^{(4\sqrt{d}\lambda^{-1} + M)\sqrt{d}h} - 1)p_\lambda(Q_{i,j})$$
$$\lesssim h^2(\lambda^{-1} + M)p_\lambda(Q_{i,j}).$$

The conclusion follows by summing over all cells $Q_{i,j}$. $\qquad\qquad\qquad\qquad\qquad\qquad\square$

We now proceed to the proof of Proposition 7. This proof would be immediate if we were working on $\mathbb{R}^d$ by combining the stability of Proposition 6 with the approximation error of Theorem 1. However, our framework in this section is that of the torus, and has to be so because there is no compactly supported measures with continuous log-densities on $\mathbb{R}^d$. In the setting of the torus, the equivalence from Eq. (4) holds for a slightly different cost function built from the heat kernel on the torus, as proved in [30] for general manifolds. This cost function is

$$\tilde{c}_\lambda(x,y) = -\lambda \log \Big( \sum_{k\in\mathbb{Z}^d} \exp \Big( -\frac{1}{2\lambda}\|x-y-k\|_2^2 \Big) \Big).$$

Let $\tilde{T}_\lambda(\mu,\nu)$ be the entropy regularized optimal transport cost as defined in Eq. (1) where the cost function $c(x,y) = \frac{1}{2}\|[x-y]\|_2^2$ is replaced by $\tilde{c}_\lambda$, and let $\tilde{S}_\lambda$ be the corresponding Sinkhorn divergence, as defined in Eq. (2). A direct extension of Theorem 1 then gives that if $\mu,\nu\in\mathcal{P}(\mathbb{R}^d)$ have bounded densities and supports then

$$|\tilde{S}_\lambda(\mu,\nu) - W_2^2(\mu,\nu)| \leq \frac{\lambda^2}{4}\max\{2I_0(\mu,\nu), I_0(\mu,\mu)+I_0(\nu,\nu)\}. \qquad (11)$$

In the next lemma, we control the error that is made when replacing $\tilde{S}_\lambda$ by $S_\lambda$, which is asymptotically exponentially small.

**Lemma 7.** *Assume that $\mu,\nu\in\mathcal{P}(\mathbb{T}^d)$ admit log-densities which are Lipschitz continuous. Then there exists $c_1, c_1', c_2 > 0$ such that*

$$0 \leq T_\lambda(\mu,\nu) - \tilde{T}_\lambda(\mu,\nu) \leq c_1 e^{-c_2/\lambda}.$$

*In particular, we have $|\tilde{S}_\lambda(\mu,\nu) - S_\lambda(\mu,\nu)| \leq c_1' e^{-c_2/\lambda}$.*

In contrast to the other statements in this paper, this one is purely asymptotic in the sense that the constants may depend on $\mu$ and $\nu$. This is due to a technical difficulty near the cut-locus where the convergence of $\tilde{c}_\lambda$ towards $c$ is only in $O(\lambda)$ which is too slow for our purposes. We can avoid this difficulty by exploiting the fact that the optimal transport map stays away from the cut locus and using the uniform convergence of the dual potentials $(u_\lambda, v_\lambda)$ towards $(u_0, v_0)$ but we are not aware of quantitative versions of these results.

*Proof.* The inequality $\tilde{T}_\lambda(\mu,\nu) \leq T_\lambda(\mu,\nu)$ is immediate since $\tilde{c}_\lambda \leq c$. The main difficulty is thus to prove the other bound. For this, let $(u_\lambda, v_\lambda)$ be the unique pair of maximizers of Eq. (9) such that $u_\lambda(0) = 0$. As $\lambda \to 0$, this pair converges uniformly to a couple of functions $(u_0, v_0)$ which is the unique solution to the unregularized dual problem such that $u_0(0) = 0$, see e.g. [5]. Letting $\tilde{F}_\lambda$ be the dual of the regularized problem Eq. (8) where $c$ is replaced by $\tilde{c}_\lambda$, we have $\frac{1}{2}\tilde{T}_\lambda(\mu,\nu) = \sup \tilde{F}_\lambda(u,v)$ where the supremum is over pairs of continuous functions on the torus. Thus we have

$$\frac{1}{2}T_\lambda(\mu,\nu) - \frac{1}{2}\tilde{T}_\lambda(\mu,\nu) \leq F_\lambda(u_\lambda, v_\lambda) - \tilde{F}_\lambda(u_\lambda, v_\lambda)$$

$$= \lambda \int_{(\mathbb{T}^d)^2} e^{(u_\lambda(x)+v_\lambda(y)-c(x,y))/\lambda}\big(e^{(c-\tilde{c})/\lambda}-1\big)\mathrm{d}\mu(x)\mathrm{d}\nu(y).$$

It remains to bound this integral and we will do so by dividing the domain $(\mathbb{T}^d)^2$ into two sets.

By the regularity theory of optimal transport on the torus [15], we know that $u_0$ is continuously differentiable (note that our assumption on the regularity of $\mu$ and $\nu$ is indeed stronger than Hölder continuity). It follows by [5, Lem. 2.4] that the optimal transport map $T$ is continuous and its graph $G = \{(x,T(x))\ ;\ x\in\mathbb{T}^d\}$ does not intersect the singular set $S$ of $(x,y)\mapsto \|[y-x]\|_2^2$, i.e. the set where this function is not differentiable. As both sets are compact, they are thus at a positive distance $2\delta > 0$ from each other. Let $G_\delta$ be the closed set of points that are at a distance less than or equal to $\delta$ from $G$ (which is itself at a distance $\delta$ from $S$). Since in our context $G$ is precisely the set of points $(x,y)$ where $u_0(x)+v_0(y) = c(x,y)$ (see again [5, Lem. 2.4]), there exists $\alpha > 0$ such that $u_0(x)+v_0(x)-c(x,y) \leq -2\alpha$ for all $(x,y)\in G_\delta^c = (\mathbb{T}^d)^2\setminus G_\delta$.

Let $(I)$ and $(II)$ be the value of the integral above on $G_\delta^c$ and $G_\delta$ respectively, so that $T_\lambda(\mu, \nu) - \tilde{T}_\lambda(\mu, \nu) \leq 2(I) + 2(II)$. On the one hand, by uniform convergence of the potentials, there exists $\lambda_0 > 0$ such that $\forall \lambda < \lambda_0$, $\|u_\lambda - u_0\|_\infty + \|v_\lambda - v_0\|_\infty \leq \alpha$ and thus $\forall \lambda \leq \lambda_0$,

$$(I) \leq \lambda e^{-\alpha/\lambda} |e^{\|c - \tilde{c}_\lambda\|_\infty / \lambda} + 1| = o(e^{-\alpha/(2\lambda)})$$

because $\tilde{c}_\lambda$ converges uniformly to $c$ as $\lambda \to 0$. On the other hand

$$(II) \leq \lambda \sup_{z \in G_\delta} (e^{(c(z) - \tilde{c}(z))/\lambda} - 1) = \lambda \sup_{z \in G_\delta} \sum_{k \in \mathbb{Z}^d \setminus \{k_0(z)\}} e^{(\|z - k_0(z)\|_2^2 - \|z - k\|_2^2)/(2\lambda)}$$

where $k_0(z)$ is such that $\|[z]\|_2 = \|z - k_0\|_2$ and is unique for $z \in G_\delta$. Letting $\beta = \inf_{z \in G_\delta, k \neq k_0(z)} \|z - k\|_2^2 - \|z - k_0(z)\|_2^2$, we have $\beta > 0$ since $G_\delta$ is at a positive distance from the singular set $S$ and we have $(II) \lesssim \lambda e^{-\beta/(2\lambda)}$ because the series $\sum_{k \neq k_0} e^{(\beta + \|z - k_0\|_2^2 - \|z - k\|_2^2)/(2\lambda)}$ is nonincreasing in $\lambda$ (notice that the exponent is nonpositive). Summing $(I)$ and $(II)$ leads to the result. $\qquad \square$

We are finally in a position to prove Proposition 7.

*Proof of Proposition 7.* We decompose the error as

$$|S_\lambda(\mu_h, \nu_h) - W_2^2(\mu, \nu)| \leq |S_\lambda(\mu_h, \nu_h) - S_\lambda(\mu, \nu)| + |S_\lambda(\mu, \nu) - \tilde{S}_\lambda(\mu, \nu)| + |\tilde{S}_\lambda(\mu, \nu) - W_2^2(\mu, \nu)|.$$

The first term is in $O(h^2(\lambda^{-1} + M + 1))$ by Proposition 6. The second term is bounded by $c_1 e^{-c_2/\lambda}$ by Lemma 7. The third term is bounded by $(\lambda^2/4) \max\{I_0(\mu, \nu), I_0(\mu, \mu) + I_0(\nu, \nu)\}$ as seen in Eq. (11), which is a variation of Theorem 1. Moreover, the assumption that $\mu$ and $\nu$ have $M$-Lipschitz continuous log-densities leads to the bound $I_0(\mu, \mu), I_0(\nu, \nu) \leq M^2$, which justifies why the statement of Proposition 7 does not requires specifically that these quantities be finite. Thus, we have

$$|S_\lambda(\mu_h, \nu_h) - W_2^2(\mu, \nu)| \lesssim h^2 \lambda^{-1} + \lambda^2.$$

Minimizing in $\lambda$ suggests to take $\lambda = h^{2/3}$ and leads to an error bound in $O(h^{4/3})$. In terms of the accuracy $\varepsilon$, we thus have $h \asymp \varepsilon^{3/4}$ and $\lambda \asymp \varepsilon^{1/2}$. The computational complexity bound follows by Proposition 2 which gives a bound in $O(n^2 \lambda^{-1} \varepsilon^{-1})$ and the fact that $n = h^{-d} \asymp \varepsilon^{-3d/4}$, hence a bound in $O(\varepsilon^{-3d/2 - 3/2})$.

For the computational complexity bound via $T_\lambda$, we use the error decomposition

$$|T_\lambda(\mu_h, \nu_h) - W_2^2(\mu, \nu)| \leq |T_\lambda(\mu_h, \nu_h) - T_0(\mu_h, \nu_h)| + |T_0(\mu_h, \nu_h) - T_0(\mu, \nu)|$$

where the first term is in $O(\lambda \log(n))$ and the second term is in $O(h)$ by Proposition 6. Thus to reach an accuracy $\varepsilon > 0$, we may choose $h \asymp \varepsilon$ and $\lambda \asymp \varepsilon / \log(n)$ which leads to a time complexity in $\tilde{O}(\varepsilon^{-2d-2})$. $\qquad \square$

# F   Analysis of the Gaussian case

Let $\mu = \mathcal{N}(a, A)$ and $\nu = \mathcal{N}(b, B)$ be Gaussian probability distributions with means $a, b \in \mathbb{R}^d$ and positive definite covariances $A, B \in \mathbb{R}^{d \times d}$. The following explicit formula for $T_\lambda$ is proven in [34]:

$$T_\lambda(\mu, \nu) = \|a - b\|_2^2 + \mathrm{tr}(A) + \mathrm{tr}(B) - 2\,\mathrm{tr}(D_\lambda^{AB}) + d\lambda(1 - \log(2\lambda)) + \lambda \log \det(2D_\lambda^{AB} + \lambda I)$$

where $A^{1/2}$ denotes the unique positive definite square root of a positive definite matrix $A$ and $D_\lambda^{AB} = (A^{1/2} B A^{1/2} + \lambda^2 I/4)^{1/2}$ (notice that $A^{1/2} B A^{1/2} = M^\top M$ for $M = B^{1/2} A^{1/2}$ is positive definite). When $\lambda = 0$, we recover the well known explicit formula (see e.g. [6]):

$$W_2^2(\mu, \nu) = \|a - b\|_2^2 + \mathrm{tr}(A) + \mathrm{tr}(B) - 2\,\mathrm{tr}(S).$$

where $S = (A^{1/2} B A^{1/2})^{1/2}$. Notice that this expression involves the squared Bures distance [6] between positive definite matrices defined as $\mathrm{d}_{\mathrm{b}}^2(A, B) \overset{\text{def.}}{=} \mathrm{tr}(A) + \mathrm{tr}(B) - 2\,\mathrm{tr}(S)$.

The expression above leads to the following formula for $\Delta = S_\lambda(\mu, \nu) - W_2^2(\mu, \nu)$:

$$\begin{aligned}
\Delta &= (\mathrm{tr}(D_\lambda^{AA}) - \mathrm{tr}(D_0^{AA})) + (\mathrm{tr}(D_\lambda^{BB}) - \mathrm{tr}(D_0^{BB})) - 2(\mathrm{tr}(D_\lambda^{AB}) - \mathrm{tr}(D_0^{AB})) \\
&\quad + \frac{\lambda}{2} \big(2 \log \det(2D_\lambda^{AB} + \lambda I) - \log \det(2D_\lambda^{AA} + \lambda I) - \log \det(2D_\lambda^{BB} + \lambda I)\big).
\end{aligned}$$

**Fourth-order expansion of $\Delta$.** Let us first expand individual terms using the fact that all the matrices involved are positive definite. We have

$$D_\lambda^{AA} = A(I + (\lambda^2/4)A^{-2})^{1/2}$$
$$= A + \frac{\lambda^2}{8}A^{-1} - \frac{\lambda^4}{128}A^{-3} + O(\lambda^5).$$

Also, since $\log\det(I + \lambda A) = \lambda\operatorname{tr}(A) - (\lambda^2/2)\operatorname{tr}(A^2) + (\lambda^3/3)\operatorname{tr}(A^3) + O(\lambda^4)$, we obtain the expansion

$$\frac{\lambda}{2}\log\det(2D_\lambda^{AA} + \lambda I) = \frac{\lambda}{2}\log\det(2A + (\lambda^2/4)A^{-1} + \lambda I + O(\lambda^4))$$
$$= \frac{\lambda}{2}\log\det(2A) + \frac{\lambda}{2}\log\det(I + (\lambda/2)A^{-1} + (\lambda^2/8)A^{-2} + O(\lambda^4))$$
$$= \frac{\lambda}{2}\log\det(2A) + \frac{\lambda^2}{4}\operatorname{tr}(A^{-1}) - \frac{\lambda^4}{96}\operatorname{tr}(A^{-3}) + O(\lambda^5).$$

Putting all pieces together with the notation $S = (A^{1/2}BA^{1/2})^{1/2}$ leads to

$$\Delta = \frac{\lambda^2}{8}\operatorname{tr}(A^{-1}) - \frac{\lambda^4}{128}\operatorname{tr}(A^{-3}) + \frac{\lambda^2}{8}\operatorname{tr}(B^{-1}) - \frac{\lambda^4}{128}\operatorname{tr}(B^{-3}) - \frac{\lambda^2}{4}\operatorname{tr}(S^{-1}) + \frac{\lambda^4}{64}\operatorname{tr}(S^{-3})$$
$$+ \lambda\log\det(2S) - \frac{\lambda}{2}\log\det(2A) - \frac{\lambda}{2}\log\det(2B)$$
$$+ \frac{\lambda^2}{2}\operatorname{tr}(S^{-1}) - \frac{\lambda^2}{4}\operatorname{tr}(A^{-1}) - \frac{\lambda^2}{4}\operatorname{tr}(B^{-1}) - \frac{\lambda^4}{48}\operatorname{tr}(S^{-3}) + \frac{\lambda^4}{96}\operatorname{tr}(A^{-3}) + \frac{\lambda^4}{96}\operatorname{tr}(B^{-3}) + O(\lambda^5).$$

The $\log\det$ terms cancel each other and some simplifications in the other terms lead to

$$\Delta = \frac{\lambda^2}{8}\big(2\operatorname{tr}(S^{-1}) - \operatorname{tr}(A^{-1}) - \operatorname{tr}(B^{-1})\big) - \frac{\lambda^4}{384}\big(2\operatorname{tr}(S^{-3}) - \operatorname{tr}(A^{-3}) - \operatorname{tr}(B^{-3})\big) + O(\lambda^5).$$

Interestingly, this expression can be expressed purely in terms of Bures distances:

$$S_\lambda(\mu,\nu) - W_2^2(\mu,\nu) = -\frac{\lambda^2}{8}\operatorname{d}_{\mathrm{b}}^2(A^{-1}, B^{-1}) + \frac{\lambda^4}{384}\operatorname{d}_{\mathrm{b}}^2(A^{-3}, B^{-3}) + O(\lambda^5).$$

This shows that the terms in this expansion are non-zero unless $A = B$ and also determines their sign.

# G  Numerical settings and additional experiments

## G.1  Sampling method

In this paragraph, we detail the setting of the *random sampling* experiments (Figure 2 and Figure 6). In those experiments, the distributions $\mu$ and $\nu$ are elliptically contoured and centered, which allows to have a closed form expression for the optimal transport cost $T_0$ and the dual potential $\varphi$ (the Lagrange multiplier associated to the first marginal constraint in the computation of $T_0(\mu,\nu)$ in Eq. (1)), which only depends on the two covariances [6]. Specifically, given two measures $\mu, \nu$ that belong to the same family of elliptically contoured distributions, with respective covariances $A$ and $B$ and with 0 means, we have

$$T_0(\mu,\nu) = \operatorname{d}_{\mathrm{b}}^2(A, B) \qquad \text{and} \qquad \varphi(x) = x^\top(\mathrm{Id} - M)x$$

where $\operatorname{d}_{\mathrm{b}}^2(A, B) = \operatorname{tr}(A) + \operatorname{tr}(B) - 2\operatorname{tr}(S)$ and $M = A^{1/2}SA^{1/2}$ where $S$ is as defined in Appendix F. Let us detail how we have chosen the covariances and our choice of elliptically contoured distribution.

**Choice of the covariances.** The covariances $A, B \in \mathbb{R}^{d\times d}$ are generated randomly, independently and identically according to the following process, that we detail for $A$. Let $M \in \mathbb{R}^{d\times k}$ be a random matrix with i.i.d. entries following a standard normal distribution $\mathcal{N}(0, 1)$, with $k = d/\alpha$ for some $\alpha \in (0, 1)$. We then define $\tilde{A} = MM^\top$, which is a random positive semidefinite matrix. By

non-asymptotic versions of the Marčenko-Pastur Theorem (e.g. [58, Eq.(1.11)]), the eigenvalues of $\tilde{A}$ are contained within a small enlargement of the interval $[(1 - \sqrt{\alpha})^2, (1 + \sqrt{\alpha})^2]$ with a high probability that increases with $d$. We then define $A = \tilde{A}/\operatorname{tr}\tilde{A}$. With our choice $\alpha = 1/3$, this allows to define generic covariance matrices of trace 1 with a controlled anisotropy: the ratio between the largest and smallest eigenvalue is with high probability of order $0.07$ for large $d$ (but note that since we work with relatively small values of $d$, this ratio is subject to fluctuations).

**Choice of the distributions.** Given a covariance $A$ we generate a sample $X$ as follows:

1. $U \sim \mathcal{U}(\mathbb{S}^{d-1})$ ( $U$ is uniformly distributed on the sphere in $\mathbb{R}^d$)
2. $Z \sim \mathcal{N}(0, 1)$
3. $R = \alpha|\arctan(Z/\beta)|^{1/d}$ where $\alpha > 0$ is such that $\mathbf{E}[R^2] = d$
4. $X = R \cdot A^{1/2} U$

Here $\beta > 0$ is a free parameter that determines the shape of the distribution and we have chosen $\beta = 2$ because it tends to yield nice bell shaped densities (see Figure 5). Also, $\alpha$ is a quantity that only depends on $d$ and $\beta$ that we estimate via Monte-Carlo integration. Let us describe the distribution of $X$.

**Proposition 14.** *The law of $X$ is elliptically contoured, centered, and has a compact support. Its covariance is $A$ and its density with respect to the Lebesgue measure (denoted by $\mu(x)$) is given by*

$$\mu(x) \propto (1 + \tan(y)^2) \exp(-\beta^2 \tan(y)^2/2) \tag{12}$$

*where $y = (\|x\|_{A^{-1}}/\alpha)^d$ and $\|x\|_{A^{-1}}^2 = x^\top A^{-1} x$. In particular, if $A$ is nonsingular then its Fisher information is finite: $I_0(\mu, \mu) < \infty$.*

It follows that if $\mu$ and $\nu$ are the densities of random variables generated via this procedure, with respective covariances $A$ and $B$, then Theorem 1 together with Proposition 1 guarantee that Proposition 4 applies. We illustrate the results of Proposition 14 in Figure 5.

*Proof.* By construction $\mu$ is elliptically contoured and centered [22, Chap. 2]. It is compactly supported because the range of $z \mapsto |\arctan(z/\beta)|$ is $[0, \pi/2)$. Also the covariance of $X$ is

$$\mathbf{E}[XX^\top] = \frac{1}{d}\mathbf{E}[R^2]A = A.$$

Let $Y = \arctan(Z/\beta)$ and let $F_Y$ (resp. $f_Y$) be the cumulative (resp. probability) distribution function of $Y$. We have for $x \in \mathbb{R}$,

$$F_R(x) = \mathbf{P}[R \le x] = \mathbf{P}[\alpha|\arctan(Z/\beta)|^{1/d} \le x] = \mathbf{P}[|Y| \le (x/\alpha)^d] = F_{|Y|}((x/\alpha)^d).$$

Differentiating this relation, it follows that $f_R(x) \propto x^{d-1} f_{|Y|}((x/\alpha)^d)$. Then by [22, Thm. 2.9 & Eq. (2.43)], we have

$$\mu(x) \propto \|x\|_{A^{-1}}^{1-d} f_R(\|x\|_{A^{-1}}) \propto f_{|Y|}((\|x\|_{A^{-1}}/\alpha)^d).$$

It thus remains to compute the density $f_{|Y|}$ which, by symmetry of $Y$ around 0, is precisely twice the density $f_Y$ for nonnegative arguments. Denoting $g(z) = \arctan(z/\beta)$, by the change of variable formula, we have

$$f_Y(y) = \frac{f_Z(g^{-1}(y))}{g'(g^{-1}(y))} \propto (1 + \tan(y)^2) \cdot \exp(-\beta^2 \tan(y)^2/2)$$

which gives the density of $\mu$, up to a multiplicative constant. Let us now show that the Fisher information $I_0(\mu, \mu) = \int_{\mathbb{R}^d} \|\frac{\nabla\mu(x)}{\mu(x)}\|_2^2 \mu(x)\mathrm{d}x$ is finite, with the assumption that $A = \operatorname{Id}$ for simplicity (the general case can be treated similarly). We have $\mu(x) = f_Y(h(\|x\|_2))$ with $h(r) = (r/\alpha)^d$ and by direct computations:

$$I_0(\mu, \mu) \propto \int_{\mathbb{R}^d} \Big(\frac{f'(h(\|x\|_2))}{f(h(\|x\|_2))}\Big)^2 \|x\|_2^{2d-2} f(h(\|x\|_2))\mathrm{d}x \propto \int_0^{\pi/2} \Big(\frac{f'(h(r))}{f(h(r))}\Big)^2 r^{3d-3} f(h(r))\mathrm{d}r$$

$$f_Y'(y) \propto \exp(-\beta^2 \tan(y)^2/2)\big(\beta^2 \tan(y)(1 + \tan(y)^2)(1 - \beta^2 \tan(y)^2)\big).$$

Figure 5: Density used for the random sampling experiments, when $A = \mathrm{Id}/d$. Left: radial profile of the density as given by Eq. (12), i.e. $t \mapsto \mu(t\vec{u})$ for some $\vec{u} \in \mathbb{S}^{d-1}$. Right: $10^4$ samples for $d = 2$.

Then by posing $z = \tan h(r)$, we get

$$I_0(\mu, \mu) \propto \int_{\mathbb{R}_+} (\beta^4 z^2 (1 - \beta^2 z^2)^2 \arctan(z)^{2-2/d}) \exp(-\beta^2 z^2 / 2) \mathrm{d}z$$

where $\propto$ in those computations just means that the right-hand side is finite if and only if the left-hand side is finite. Since the right-hand side is finite, this shows that $I_0(\mu, \mu) < \infty$. $\qquad\square$

### G.2  Additional random sampling experiment

On Figure 6, we show the same experiment as in Section 5 but in dimension $d = 10$ and moreover we report the error on the transport cost $T_0(\mu, \nu)$ and the rate of Theorem 2, which were not shown on Figure 2. The plot on the right shows the estimation error on $T_0(\mu, \nu)$, which is the quantity that we control in our theoretical analysis. This plot confirms several of our results: (i) the convergence rate in $n^{-2/d}$ of the plug-in estimator proved in Theorem 2 (note that we compute it with a small entropic regularization, which might explain the slight deviation from the rate $n^{-2/d}$ that we observe for $n$ large), and (ii) the fact that $T_\lambda$ has a much larger bias than $S_\lambda$ and $R_\lambda$. Even more interestingly, $S_\lambda$ and $R_\lambda$ have a smaller error than the plug-in estimator. However, we should also be cautious when interpreting such a plot because $T_0(\mu, \nu)$ is a scalar, and it is easy to make the error vanish when varying a parameter, such as $n$ or $\lambda$. In particular, the local minimum observed for $S_\lambda$ and $R_\lambda$ is simply due to the fact that the error changes its sign as $n$ grows.

This phenomenon led us to report the error on a different quantity, the $L^1$ error on the potential, which is not subject to this phenomenon and which also raises interesting open questions. Notice however that this quantity may behave quite differently than the estimation error on $T_0(\mu, \nu)$. In particular, we see on Figure 5-(left), that the rate of convergence of the plug-in estimator is in fact faster than $n^{-2/d}$ in this experiment.

### G.3  Additional figures for the discretization experiment

Figure 7 shows the same setting as on Figure 4 and gives more details. The densities of $\mu$ and $\nu$ on the 1-dimensional torus $\mathbb{T}$ are shown on the top row at several levels of discretization. The two other rows show the evolution of the estimated potentials as $n$ varies for the optimal $\lambda$ (middle row) or as $\lambda$ varies for $n$ large (bottom row) towards the true potentials $(u_0, v_0)$ (shown in dark color). Here $u_0$ is the Lagrange multiplier associated to the first marginal constraint in the computation of $T_0(\mu, \nu)$ in Eq. (1) and $v_0$ is the one associated to the second marginal constraint. On Figure 7, we denote by $(u_h, v_h)$ the potentials associated to the estimator $T_\lambda$ and by $(\bar{u}_h, \bar{v}_h)$ those associated to the estimator $S_\lambda$, as defined in Section 5. This figure illustrates that for $\lambda$ large, the error is systematically smaller with the debiasing terms.

Figure 6: $L^1$ error on the first potential (left) and error on the estimated cost (right) for different estimators, for $\mu, \nu$ smooth compactly supported distributions with $d = 10$, as a function of $n$ for $\lambda = 1$. Error bars show the standard deviation on 30 realizations

Figure 7: Rows 1 and 2: convergence of the dual potentials $(u_{0,h}, v_{0,h})$ and $(\bar{u}_{0,h}, \bar{v}_{0,h})$ towards $(u_0, v_0)$ for decreasing sampling step $h$. The top row shows the discretized measures $(\mu_h, \nu_h)$ (the measure is a sum of Dirac masses, which is vizualized as a piecewise constant function to indicate the cells over which the densities have been integrated). Last row: same but for the convergence of $(u_{\lambda,0}, v_{\lambda,0})$ and $(\bar{u}_{\lambda,0}, \bar{v}_{\lambda,0})$ as $\lambda$ gets smaller.