[Reviews · NeurIPS 2020]

Review 1

Summary and Contributions: This paper performs an extensive analysis of the Sinkhorn divergence as an estimator for the squared Wasserstein distance. They provide a sample complexity bound in the case when the source and target distribution are compactly supported, and show that it infers computational advantages due to a reduced need for regularization. They also improve on sample complexity bounds for the plug-in estimator (for source and target not equal). Lastly, they also formulate a Richardson extrapolation variant of the estimator, and bound the Sinkhorn estimator error in terms of the Fisher information.

Strengths: The paper seems to provide strong theoretical results that will be of great interest to those interested in computational complexity and statistical power of Sinkhorn divergences as estimators.

Weaknesses: Perhaps the main weakness of the work might be its relevance to more applied OT practitioners. It lends some theoretical support to methods which utilize Sinkhorn divergence, but it's not clear to me that it leads to any real practical gain.

Correctness: I was not able to check all the proofs, due to its very technical nature, but what I did check was correct. For the empirical methodology, I still felt like it might have been interesting to see the plots for the error in squared Wasserstein distance. As acknowledged, none of the theory applies to the L1 dual potential error, so I wasn't sure exactly how much faith to put in the empirical support that they provide.

Clarity: Yes, I found it quite well written.

Relation to Prior Work: The authors do cite Mena & Weed, and Genevay et al., but it might be helpful to discuss the relation of the current sample complexity results with these previous bounds .

Reproducibility: Yes

Additional Feedback: EDIT AFTER AUTHOR FEEDBACK: Thank you to the authors for responding to my concerns. I was already positive on the work and will be leaving my score as is.


Review 2

Summary and Contributions: The paper introduces a faster algorithm to compute the squared Wasserstein distance. Given two measures, \mu and \nu over the d-dimensional Euclidean space, the Wasserstein distance between the measures is the integral of the squared Euclidean distance with respect to the best possible coupling of \mu and \nu. The authors propose two estimators approximating the squared Wasserstein distance. They additionally show that under the right assumptions, these estimators yield good approximation algorithms for the squared Wasserstein distance, which are computationally faster than the previously studied ones.

Strengths: The authors introduce the problem studied in the paper in a very clear and coherent manner. The estimators are well presented, and the high level description of both the problem as well as the solution and the structure of the result are very well written.

Weaknesses: The vast majority of proofs are not included in the paper, and the formal foundation of the results is not clearly presented. It seems like the paper is not self-contained, and that most of the merit is in the supplementary material.

Correctness: Correctness was very difficult to verify, as most of the proofs are in the supplementary material.

Clarity: The introduction is very well written.

Relation to Prior Work: Yes.

Reproducibility: Yes

Additional Feedback: The authors refer (e.g. lines 74,113) to appendices instead of the full/supplementary version. The submission should be self-contained. General remark: I think that the authors tried to include too many results in the paper, which made it impossible to present any of them properly. I would try to focus on one or two results, and give a full(er) presentation. I realize that the author's guideline permits omitting the proofs, and that is indeed perfectly fine. However, as I see it, the paper in its final published version must be self contained, and it should be possible to read it as a whole. When introducing a theoretical result, there should be at least a high-level description/explanation, convincing the reader that the theory is solid. In my opinion, that is not the case for this paper.


Review 3

Summary and Contributions: This paper propose and analyze estimates of the squared Wasserstein distance by Sinkhorn divergence and its Richardson extrapolation based on entropic regularization. Compared to the plug-in estimator, the authors show that with comparable sample complexity, the estimate by Sinkhorn divergence allows higher regularization levels, thus leading to a speedup of the estimate in computational complexity to achieve the same biasing/approximation error. Moreover, under the assumption of the regularity of the approximation error, the authors proves that the Richardson extrapolation has improved statistical and computational efficiency.

Strengths: The strengths of the paper are: 1. improved estimate of the upper bound for the plug-in estimator; 2. rigorous theoretical justification of the advantages in computational complexity and approximation error for the estimates by Sinkhorn divergence and its Richardson extrapolation; 3. interesting numerical demonstration of the improvement of the proposed estimates with respect to the regularization parameter.

Weaknesses: It is not clear how to systematically and optimally choose/compute the regularization parameter \lambda^* to achieve the more accurate and faster approximation of the Wasserstein distance, and if optimizing the parameter with additional cost would significantly compromise the total computational complexity. Can the author verify the convergence rates with respect to the number of samples in Figure 1 by adding a plot for the theoretical rates? Can the author be more specific about the smoothness of the problem, in terms of the bound for the derivatives and its relation in the approximation error estimates?

Correctness: The dynamical formulation (4) does not seem to be correct without stating the relation between \rho(t, x) and \mu and \nu.

Clarity: The paper is well written and easy to follow. However, I did not check the proofs.

Relation to Prior Work: The discussion on how this work differs from previous contributions is clear.

Reproducibility: Yes

Additional Feedback:


Review 4

Summary and Contributions: The Wasserstein square distance is emerging as an important metric to compute the similarity between two probability distributions. It has advantages over KL-divergence especially when the distributions have disjoint support. In practice there is a famous Sinkhorn algorithm which essentially takes the set of two sets of points (one from each distributions) and solves an LP problem but with an entropy regularization term. The entropy regularization is mostly to speed up the solution where the regularized LP problem can be solved by a form of alternating maximization. However the regularization parameter \lamda biases the estimator. The proposed paper uses to estimators (Equation 2 and 3) based on the Sinkhorn divergence (S_\lambda) shows that the estimators have comparable sample efficiency but better computational efficiency.

Strengths: The paper makes a theoretical contribution where it proposes the Sinkhorn divergence and its extension (Equation 3) to estimate the Wassertein distance. Sinkorn divergence is defined in terms of T_\lambda (Equation 1). The Sinkhorn algorithm directly estimates T_\lambda and the authors show theoretically that by using the Sinkhorn divergence comparable accuracy can be achieved for higher values of \lambda. In Proposition 7 the claim for T_\lambda is \lambda = epsilon/log(n) but for S_\lambda the claims is \lambda = epsilon^2. The running time of T_\lambda is inversely proportional to \lambda. Figure 1(b) shows the advantage of S_\lambda. For example at \lambda = 1, T_\lambda error is substantially higher than the S_\lambda error

Weaknesses: The authors have kept d=5 in the experiments. Since Wasserstein-GANS and Wasserstein Autoencoders are examples where Wasserstein distances are being used, it would be good to see empirical errors as a function of dimensionality.

Correctness: I did not go through the proofs of the claims but the claims appear credible.

Clarity: For a theory paper it is well-written and both the strengths and weaknesses of the theory are clearly explained.

Relation to Prior Work: Yes, prior work is reasonably well-explained.

Reproducibility: Yes

Additional Feedback:


Review 5

Summary and Contributions: This paper discusses the effectiveness of estimating the squared Wasserstein distance by Sinkhorn divergence in terms of sample complexity and computational complexity. Their argument consists of the following new results: 1. Bounds of the difference between Sinkhorn divergence and Wasserstein distance according to the value of \lambda and Fisher information. 2. Evaluation of sample complexity of the plug-in estimator using empirical distribution. 3. Evaluation of sample complexity and computational complexity of the Sinkhorn divergence-based estimator and the traditionally used entropic regularization-based estimator. Based on these results, the authors conclude that the Sinkhorn divergence-based estimator is superior to the entropic regularization-based estimator. Furthermore, they propose a new estimator by combining the Sinkhorn divergence-based estimator and Richardson extrapolation. The effectiveness of the proposed methods is confirmed by numerical experiments.

Strengths: 1. The authors propose a new and effective method for the problem of squared Wasserstein distance estimation, which is considered to be very important in machine learning community. 2. Detailed theoretical evaluations on the complexities of various estimators. These results are important not only because they support the effectiveness of the proposed methods, but also because they are expected to be used in the analysis of various methods in the future. 3. The structure of the paper is well designed, and the propositions and references necessary are clearly shown, so it is easy to follow the discussion and understand the contents.

Weaknesses: 1. In experimental results, authors measure errors by L1 error, which does not appear in theoretical results. The reason for this is understandable, but this makes the results not interpretable and hard to determine whether the experiments verify their claims. 2. Computational time is not evaluated enough in the experimental section.

Correctness: Although I do not check details of the proofs, this paper appears technically sound.

Clarity: The paper is well-structured and easy to follow.

Relation to Prior Work: The relationships between this work and previous works are clearly discussed in "Previous Works" subsection in section 1.

Reproducibility: Yes

Additional Feedback: This paper is well written and has significant results. Several comments and questions: 1. I would like to see experimental results about the relationship between error and dimension d because d appears most of the theoretical results. Is it difficult? 2. I felt that the experimental results show the effectiveness of Richardson is not so significant. Why? If d were larger, would it be more effective? --- Comments aftter rebuttal phase I read the author's response and believe that the promised updates will improve the quality of the paper. I will keep my score.

[Author Response · NeurIPS 2020]

# Response to reviewers comments for the paper: "Faster Wasserstein Distance Estimation with the Sinkhorn Divergence"

We thank the reviewers for their comments and suggestions. Hereafter, we list reviewers' (paraphrased) comments (**C**) followed by our responses (**R**). These answers will translate into clarifications in the final version of the paper.

## Response to reviewer #1's comments

- **(C)** *It's not clear to me that the paper leads to any real practical gain.* **(R)** Our goal is mostly to deepen our theoretical understanding of known objects, such as the Sinkhorn divergence.
- **(C)** *It might have been interesting to see the plots for the error in squared Wasserstein distance.* **(R)** We will move such plots (currently in the supplementary material) to the main paper.
- **(C)** *The authors do cite Mena & Weed, and Genevay et al., but it might be helpful to discuss the relation of the current sample complexity results with these previous bounds .* **(R)** Those works were concerned with estimating the *regularized* optimal transport cost, while our goal is to estimate the *unregularized* cost. These are very different problems: the latter is cursed by the dimensionality while the former is not.

## Response to reviewer #2's comments

- **(C)** *The submission should be self-contained.* **(R)** Our submission is self-contained. The main paper contains all our results, with a self-contained narrative, and all the proofs are provided in the supplementary material, as referenced in the main text. We have often included proof ideas in the main text. This is consistent with NeurIPS guidelines which state that "Authors may submit up to 100MB of supplementary material, such as appendices, proofs, derivations [...]".
- **(C)** *I would try to focus on one or two results, and give a full(er) presentation..* **(R)** We think that the set of results that we present form a coherent story. We will use an additional page in the final version to give more details and ease the reading.

## Response to reviewer #3's comments

- **(C)** *How to choose the best regularization parameter $\lambda^*$?* **(R)** The optimal value $\lambda^*$ depends on quantities which are typically not known so our theory is not yet able to answer this question. In practice, many machine learning tasks involving the Wasserstein distance come with a performance criterion, in which case cross-validation can be used.
- **(C)** *Can we verify the convergence rates with respect to the number of samples in Figure 1 by adding a plot for the theoretical rates?* **(R)** As mentioned in the text, we do not know the theoretical rates for the quantity shown in Figure 1 ($L^1$ error on the potential), so we cannot plot it. Note that the observed rate is different from $n^{-2/d}$.
- **(C)** *Be more specific about the required smoothness of the problem, in terms of the bound for the derivatives.* **(R)** This is a question that Proposition 1 partially addresses. We will add details on how to guarantee smoothness of the Brenier potential assuming smoothness of the densities (a typical and difficult problem in optimal transport theory).

## Response to reviewer #5's comments

- **(C)** *It would be good to see empirical errors as a function of dimensionality [also asked by **reviewer #6**].* **(R)** Such plots would not be very interpretable since we did not track the full dependency in the dimension $d$ in the theory. Our focus is on the rates in the sample size $n$. Plots in higher dimension are qualitatively similar, but with different slopes.

## Response to reviewer #6's comments

- **(C)** *It is hard to determine whether the experiments verify the claims.* **(R)** As justified in the main text, Figure 1 and 2 are not intended to verify our theorems but rather to exhibit related phenomena and suggest future research directions. We will add in the main paper a plot for the error in cost and a plot to illustrate Theorem 2, which verify our claims.
- **(C)** *Computational time is not evaluated enough in the experimental section.* **(R)** We will expand a bit and add a plot.
- **(C)** *In the experimental results, the effectiveness of Richardson is not so significant. Why?* **(R)** We observed a significant debiasing effect (see Figure 1-(a)), but indeed no clear statistical or computational gain. We believe that this is mainly due to "constants" in the bounds: compared to $S_\lambda$, the estimation error for $R_\lambda$ is up to 3 times larger and the computational time is up to 3 times larger.

[Meta-Review · NeurIPS 2020]

Most reviewers agreed that the paper provides novel and solid theoretical results on the properties of the Sinkhorn divergence for estimating the Wasserstein distance. Some reviewers have suggested some minor points that can be improved (eg plots of error, adding computational time analysis, ...) but overall, they are happy to have this paper at Neurips.